# Integrative functional genomics identifies regulatory mechanisms at coronary artery disease loci

Clint L. Miller[1,*], Milos Pjanic[1,*], Ting Wang[1], Trieu Nguyen[1], Ariella Cohain[2], Jonathan D. Lee[1,3], Ljubica Perisic[4], Ulf Hedin[4], Ramendra K. Kundu[1], Deshna Majmudar[1], Juyong B. Kim[1], Oliver Wang[1], Christer Betsholtz[5,6], Arno Ruusalepp[7,8], Oscar Franzén[2,8], Themistocles L. Assimes[1], Stephen B. Montgomery[3,9], Eric E. Schadt[2], Johan L.M. Björkegren[2,6,10] & Thomas Quertermous[1]

Coronary artery disease (CAD) is the leading cause of mortality and morbidity, driven by both genetic and environmental risk factors. Meta-analyses of genome-wide association studies have identified >150 loci associated with CAD and myocardial infarction susceptibility in humans. A majority of these variants reside in non-coding regions and are co-inherited with hundreds of candidate regulatory variants, presenting a challenge to elucidate their functions. Herein, we use integrative genomic, epigenomic and transcriptomic profiling of perturbed human coronary artery smooth muscle cells and tissues to begin to identify causal regulatory variation and mechanisms responsible for CAD associations. Using these genome-wide maps, we prioritize 64 candidate variants and perform allele-specific binding and expression analyses at seven top candidate loci: 9p21.3, *SMAD3*, *PDGFD*, *IL6R*, *BMP1*, *CCDC97/TGFB1* and *LMOD1*. We validate our findings in expression quantitative trait loci cohorts, which together reveal new links between CAD associations and regulatory function in the appropriate disease context.

[1] Department of Medicine, Division of Cardiovascular Medicine, Stanford University School of Medicine, Stanford, California 94305, USA. [2] Department of Genetics and Genomic Sciences, Icahn Institute for Genomics and Multiscale Biology, Icahn School of Medicine at Mount Sinai, New York, New York 10029, USA. [3] Department of Genetics, Stanford University School of Medicine, Stanford, California 94305, USA. [4] Department of Molecular Medicine and Surgery, Karolinska Institutet, Stockholm SE-171 77, Sweden. [5] Department of Immunology, Genetics and Pathology, Rudbeck Laboratory, Uppsala University, Uppsala SE-751 05, Sweden. [6] Department of Medical Biochemistry and Biophysics, Vascular Biology Unit, Karolinska Institutet, Stockholm SE-171 77, Sweden. [7] Department of Cardiac Surgery, Tartu University Hospital, Tartu 50406, Estonia. [8] Clinical Gene Networks AB, Stockholm SE-114 44, Sweden. [9] Department of Pathology, Stanford University School of Medicine, Stanford, California 94305, USA. [10] Department of Physiology, Institute of Biomedicine and Translation Medicine, University of Tartu, Tartu 50406, Estonia. * These authors contributed equally to this work. Correspondence and requests for materials should be addressed to C.L.M. (email: clintm@stanford.edu) or to T.Q. (email: tomq1@stanford.edu).

Coronary artery disease (CAD) remains the leading cause of mortality and morbidity in the world, despite advances in treatment and lifestyle modification. As a complex disease, both genetic and environmental factors contribute to cumulative disease risk across human populations[1,2]. Meta-analyses of genome-wide association studies (GWASs) through the CARDIoGRAMplusC4D consortium have now identified 152 susceptibility loci for CAD[3,4], which explain ∼10% of the estimated heritable risk. More recent fine-mapping analyses of the CARDIoGRAMplusC4D consortium using phased haplotypes from the 1000 Genomes Project have revealed an additional 10 CAD loci[5]. These analyses have shed light on a number of biologically relevant pathways involving genes that appear to be operating in the vessel wall, independent of classical risk factors[6,7].

The majority of variants identified through GWAS (including those in linkage disequilibrium (LD)) represent common single-nucleotide polymorphisms (SNPs) located outside protein-coding sequences, which are predicted to function via cis- or trans-regulatory changes in gene expression[8]. A number of studies have experimentally validated these effects at individual loci for complex diseases, including CAD[9–11], cancer[12], metabolic disorders[13] and blood disorders[14]. However, a need exists for more scalable and sensitive approaches to detect causal variants and the underlying mechanisms for multiple loci. For instance, expression quantitative trait (eQTL) mapping and allelic expression imbalance (AEI) have been used extensively in lymphoblastoid cell lines, monocytes and other cells to identify regulatory variants[15–17]. Similarly, assays that measure chromatin accessibility and transcription factor (TF) binding have been critical for prioritizing regulatory variants[18–20]. Nonetheless, it is now clear that integrative and dynamic multi-omic analyses in primary cells/tissues may be necessary to disentangle the mechanisms of regulatory variants under different disease microenvironments[21–23].

Human coronary artery smooth muscle cells (HCASMCs) constitute the majority of the vessel wall and via their contractile functions are largely responsible for distributing blood to the heart muscle. However, these cells also undergo phenotypic modulation or 'epigenetic re-programming' to a highly proliferative and invasive state in response to vessel injury, changes in blood flow, and during lesion expansion with disease[24]. During atherosclerosis (hardening of the artery wall), these cells secrete excessive extracellular matrix, undergo apoptosis, and contribute to and remodel the surrounding fibrous cap. Recent lineage tracing studies in mice have convincingly demonstrated that up to 80% of the lesion cells (including mesenchymal stem cells and macrophage-like cells) are smooth muscle cell (SMC) derived[25]. Thus, we hypothesized that investigating the epigenetic mechanisms of gene regulation in HCASMCs may provide greater insights into disease associations and the underlying biology of the vessel wall.

Variation in chromatin accessibility and TF binding is a dominant mechanism of variation in gene expression[26], given more than half of eQTLs in 70 lymphoblastoid cell lines overlap with accessible DNaseI hypersensitivity sites (DHSs)[27]. Recently developed methods to interrogate chromatin accessibility include the Assay for Transposase Accessible Chromatin (ATAC-seq), which has an advantage over other methods in that it requires a fraction of the starting material to simultaneously assess chromatin state, nucleosome profiles and TF footprints[28]. Here we employ ATAC-seq to generate epigenomic profiles in primary cultured HCASMCs stimulated with various growth factors, as well as in normal and atherosclerotic human coronary artery tissues. We integrate these data with chromatin immunoprecipitation-sequencing (ChIP-seq) profiles for TF binding and the active enhancer histone modification H3K27ac to define HCASMC-enriched cis-regulatory mechanisms (Supplementary Fig. 1). We incorporate publicly available annotations through ENCODE, Epigenomics Roadmap and eQTL databases to identify and experimentally validate seven loci using allele-specific binding and expression analyses in HCASMCs. Last, we leverage cis-eQTL analyses in large external cohorts of normal and atherosclerotic arteries to validate our approach ex vivo. Together, these multi-dimensional data further advance our understanding of non-coding regulatory variation in a complex disease, such as CAD.

## Results

**Chromatin accessibility in perturbed HCASMCs identifies TFBS**. The vast majority of epigenomic data in public databases are derived from cells cultured under static conditions, which may not reflect the native and disease-related cellular signalling environments[22,29]. To mimic the phenotypic plasticity of vascular smooth muscle cells during atherogenesis[30,31], we treated quiescent HCASMCs under serum-free control, transforming growth factor-β (TGF-β; pro-differentiation phenotype), platelet-derived growth factor-BB or DD (PDGF-BB or PDGF-DD; de-differentiation phenotype) conditions for 6 h and profiled the epigenome using deep ATAC-seq. These upstream growth factors were previously shown to activate changes in gene expression and chromatin remodelling at the TCF21 locus[10], and are enriched at CAD loci overall (Supplementary Data 1). We obtained an average of 100 million Tn5-integrated mapped reads (200 million paired) and average detection of ∼150,000 open chromatin peaks per sample, which resulted in high signal-to-noise ratios (Fig. 1a; Supplementary Fig. 2a–c), low individual sample variability ($r^2 = 0.94$ biological replicates; Fig. 1b), and robust CTCF centred footprints and resolved nucleosome profiles (Supplementary Fig. 3). To identify relevant transcription factor-binding sites (TFBSs), we performed de novo motif enrichment analysis of each stimulated condition using serum-free control as a background. The top de novo motifs in TGF-β-treated HCASMCs included the AP-1 family members (for example, ATF3), followed by TGF-β-related TFs (TEAD1/4, SMAD2 and RUNX1; Fig. 1c). In contrast, in PDGF-BB-treated HCASMCs, we identified even more enriched AP-1 motifs (for example, FRA1), followed by CTCF (chromatin looping factor), ETS1 (pro-proliferative factor) and NFY (potent repressor of endothelial marker, von Willebrand Factor; Fig. 1d). Hierarchical clustering of open chromatin regions overlapping CAD loci revealed similarities within each donor sample and the nearest annotated genes (Fig. 1e), despite observed differences between stimulations (Supplementary Fig. 2). Functional enrichment analysis of stimulated open chromatin regions overlapping the GWAS catalogue showed significant enrichment for heart disease, atherosclerosis and related Disease Ontology terms (Fig. 1f; Supplementary Data 2). We also generated functional association networks of CAD-associated genes in TGF-β-stimulated open chromatin regions, which revealed SMAD3 as one of the most connected CAD genes, providing a link to TGF-β signalling and SMC functions in the vessel wall (Supplementary Fig. 2d–e; Supplementary Data 2 and 3).

**Integrative epigenomic profiling prioritizes CAD variants**. To prioritize all CAD variants, we first combined a candidate list from CARDIoGRAMplusC4D replicated loci and those associated at false discovery rate (FDR) of 5%, as well as novel loci identified from CARDIoGRAMplusC4D 1000G fine-mapping efforts[5]. To capture causal SNPs that are in high LD with the lead

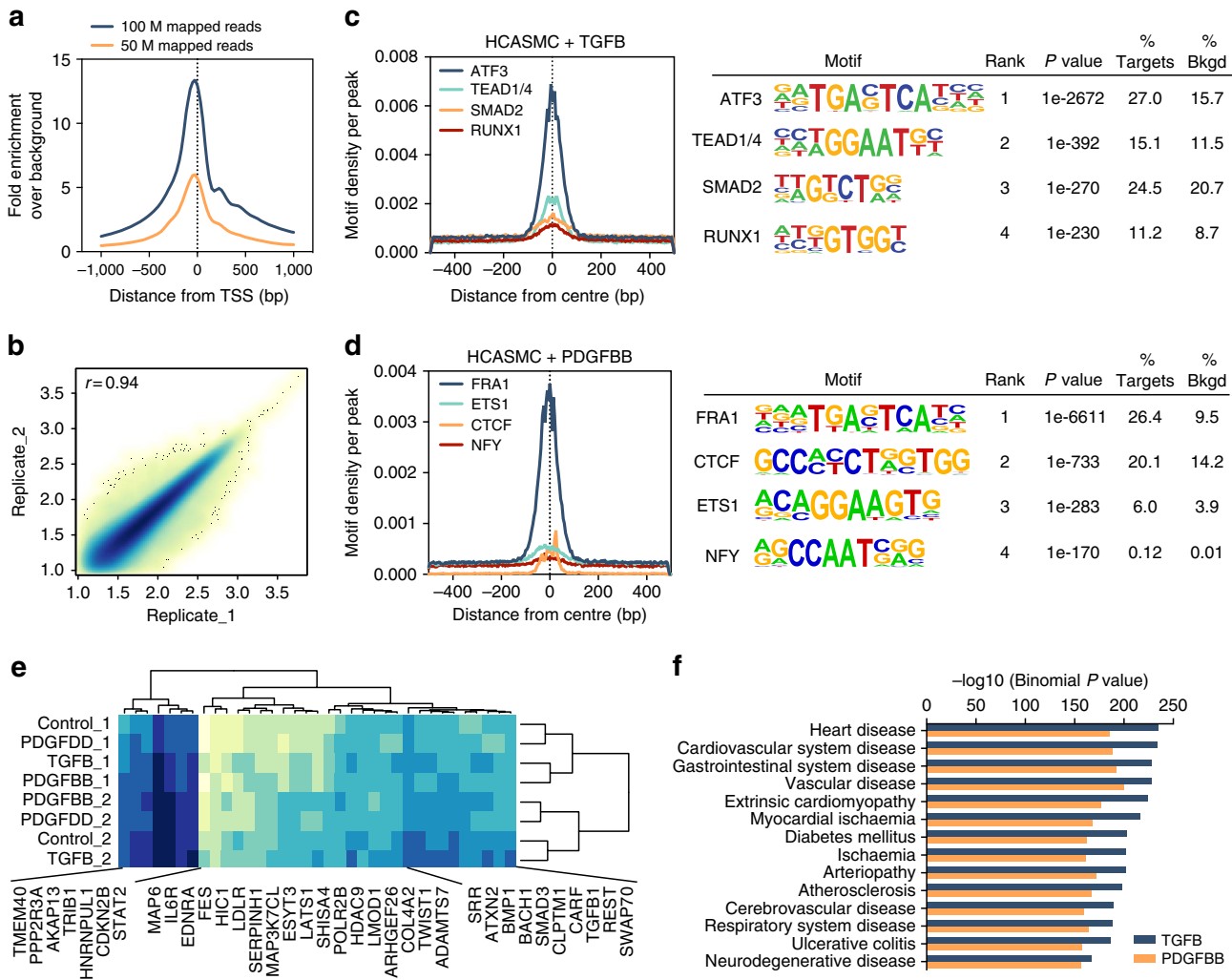

**Figure 1 | Chromatin accessibility in perturbed HCASMC identifies AP-1 and state-dependent TFBS.** (**a**) ATAC-seq signal-to-noise ratio, calculated as distribution of ATAC-seq reads centred on TSS in a 1,000-bp window, normalized to total mapped reads and comparing 50 and 100 million mapped reads from representative data sets ($n = 22$ biological replicates). (**b**) Scatter plot showing correlation of ATAC-seq tag intensities from two independent biological replicates ($r^2 \sim 0.94$). (**c**) Histogram distribution and resulting $P$ value (via cumulative binomal distribution) for top four enriched motifs identified using de novo motif enrichment analysis in open chromatin peaks in HCASMCs treated with TGF-β1 ($n = 2$ biological replicates per condition). (**d**) Similar results shown above for HCASMCs treated with PDGF-BB. (**e**) Hierarchical clustering of open chromatin normalized read counts in stimulated HCASMCs from two independent donors (1 and 2), with clustering on CAD loci annotated to transcription start site of nearest gene. (**f**) Genomic Regions Enrichment of Annotations Tool (GREAT) analysis of stimulated HCASMC open chromatin regions overlapping entire GWAS catalogue ($n = 2$ biological replicates), showing enrichment for Disease Ontologies relative to whole-genome background. $P$ values were calculated using a combination of binomial and hypergeometric tests.

SNPs, we applied a correlation cutoff of $r^2 \geq 0.8$ in Europeans using 1000 Genomes Phase 1 data (http://browser.1000genomes.org/). This resulted in 5,240 total candidate CAD regulatory SNPs present in the human genome (Supplementary Data 4). To rule out the possibility that these candidate SNPs are linked to potentially damaging protein-coding SNPs, we ran these through PolyPhen-2 (http://genetics.bwh.harvard.edu/pph2/) and SIFT (http://sift.jcvi.org), which identified eight and six non-synonymous missense/nonsense variants, respectively, predicted to be damaging (Supplementary Data 5 and 6). One of these missense variants, rs867186 (Ser219Gly), in *PROCR1* has been previously reported to explain variable levels of soluble endothelial protein C receptor associated with thrombosis and CAD risk[32]. Another missense variant, rs11556924 (His363Arg), in *ZC3HC1* may alter cell cycle entry and was associated with carotid intima-media thickness in rheumatoid arthritis (RA) patients[33]. Nonetheless, this suggests that the majority of the

remaining variants are likely impacting regulatory elements. We hypothesized that integrating multiple genome-wide regulatory features in HCASMCs would help further prioritize these variants into relevant pathways.

Given that open chromatin regions alone may not delineate functional SNPs, we also performed ChIP-seq in HCASMCs for the histone modification H3K27ac, which has been shown to detect regulatory variants in active enhancer regions[34]. Interestingly, from the H3K27ac data 16 CAD SNPs were found to overlap four super-enhancer regions, potentially indicating an additional regulatory mechanism in CAD (Supplementary Fig. 4). Also, to begin to study *cis-* and *trans-*acting causal transcription mechanisms, we restricted our functional mapping analysis to those variants located in validated binding sites by incorporating HCASMC ChIP-seq peaks for relevant TFs including, JUN, JUND and TCF21 (ref. 35). We have previously shown that TCF21 and AP-1 TFs

bind proximal to CAD loci, and TCF21 target regions are enriched for CAD variants in LD with causal variants[10,35]. The results of overlapping CAD SNPs in these data sets were $n = 323$, $n = 462$ and $n = 193$, for open chromatin, active enhancers and TF binding, respectively (Fig. 2a). The combined overlap resulted in 87 candidate CAD SNPs at 11 loci (2.45 odds ratio; Fig. 2b; Supplementary Fig. 4), suggesting that these are more likely to be functional SNPs in HCASMCs. As a comparison, we overlapped these data sets with the entire GWAS catalogue (including SNPs in LD at $r^2 \geq 0.8$) or other chronic inflammatory diseases (ulcerative colitis (UC) and inflammatory bowel disease (IBD)), which resulted in 1.4, 1.8 or 1.9 odds ratios of enrichment (Fig. 2b; Supplementary Fig. 5). Overall, incorporating TCF21 peaks improved the enrichment compared with JUN peaks, or when combined with ATAC open chromatin or H3K27ac peaks (Fig. 2b; Supplementary Fig. 5). ATAC-seq open chromatin regions centred on all GWAS SNPs displayed a local peak of enrichment (Fig. 2c). We further observed open chromatin

regions centred on CAD-associated variants to display a broad pattern of enrichment, distinct from those which are directly centred on CTCF motifs, yet strikingly similar to those in binding peaks for TCF21 and JUN (Fig. 2d). We then plotted the significance of SNPs overlapping open chromatin relative to their fold enrichment for all GWAS variants using a cumulative binomial distribution test (Fig. 2e). This revealed the CARDIOGRAMplusC4D associated SNPs to cluster with other highly significant ($-\log P$ value $\sim 75$) and enriched ($>10$-fold) SNPs for diseases with cardiovascular and chronic inflammatory (for example, UC, Crohn's disease, RA and so on) origin and unexpected overlap with bone mineral density and Schizophrenia SNPs, which appears to be cell-type dependent (Supplementary Figs 6 and 7). It is possible that the chronic inflammatory component of CAD may drive the majority of the strong enrichment signals of UC, Crohn's disease, RA, as well as myocardial infarction and insulin resistance, reflecting shared genetic aetiology.

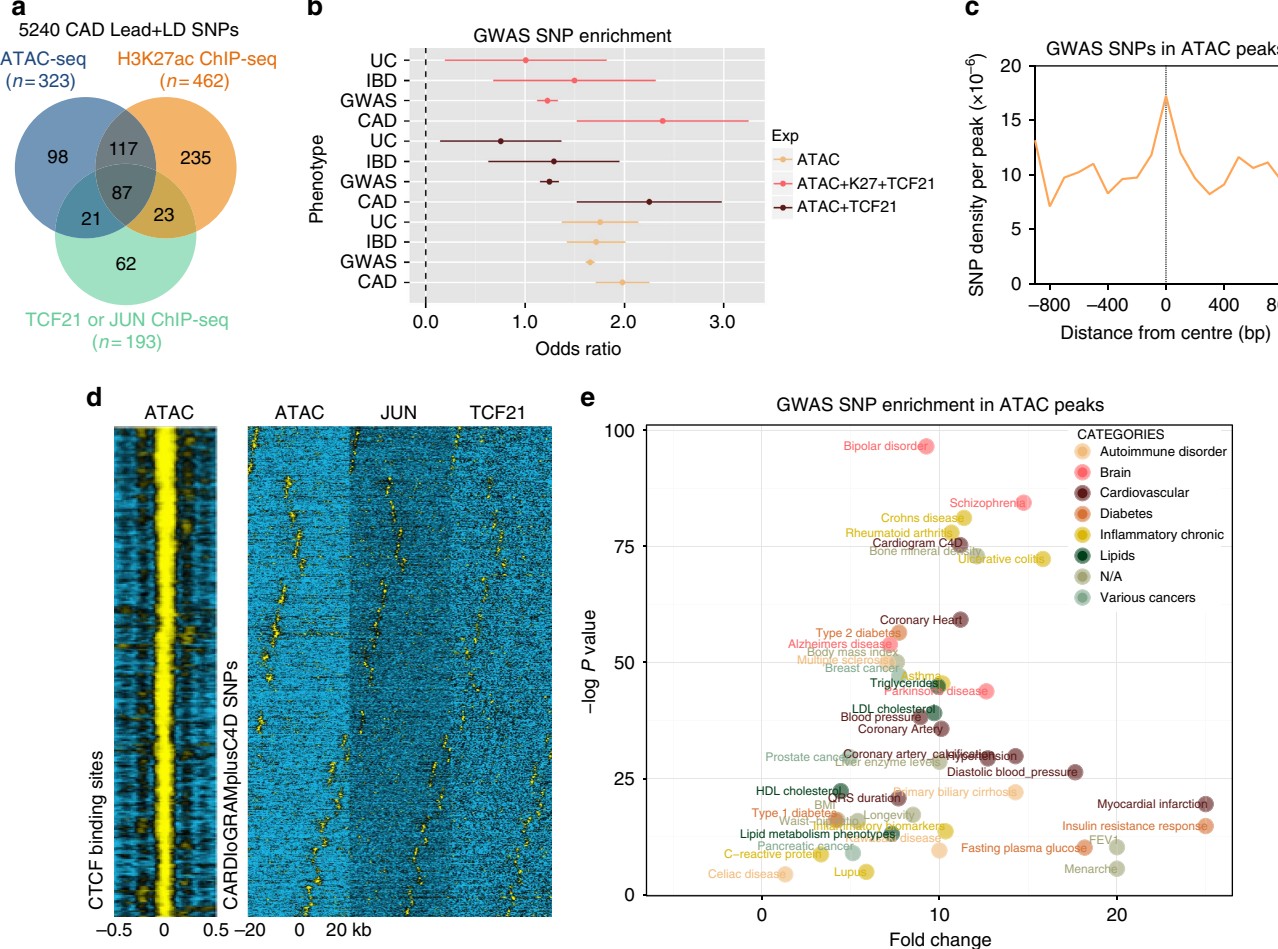

**Figure 2 | Integrative epigenomic profiling in HCASMC prioritizes CAD regulatory variants.** (**a**) Venn diagram of overlapping 5240 candidate CAD-associated variants (including those in high linkage disequilibrium at $r^2 \geq 0.8$) with HCASMC ATAC-seq open chromatin regions ($n = 323$), H3K27ac ChIP-seq active enhancer regions ($n = 462$) or TF binding via TCF21 or AP-1 ChIP-seq ($n = 193$). Unique overlapping numbers shown for combined overlaps, respectively. (**b**) Forest plot depicting odds ratio (OR) of enrichment for CARDIoGRAMplusC4D (CAD), inflammatory bowel disease (IBD), ulcerative colitis (UC) or entire GWAS catalogue SNPs in individual or combined HCASMC data sets as calculated using the Fisher's exact test. Dots represent mean OR and lateral lines represent 95% confidence intervals. (**c**) Histogram distribution of globally normalized GWAS SNPs in regions centred on HCASMC open chromatin regions within a 1-kb window. (**d**) Heatmap distribution of HCASMC open chromatin regions centred on CTCF motif (from JASPAR) within a 0.5-kb window (left panel). Hierarchical clustering heatmap showing distribution of ATAC-seq open chromatin, JUN or TCF21 ChIP-seq binding regions centred on 5,240 CARDIoGRAMplusC4D SNPs (right panels). (**e**) Two-dimensional scatter plot of GWAS SNPs in HCASMC open chromatin regions showing most significant enrichment for cardiovascular (CARDIoGRAMplusC4D and coronary heart), brain and autoimmune phenotypes in upper right quadrant. Data shown are representative of $n = 10$ biological replicates in HCASMCs cultured under normal growth conditions.

***Ex vivo* chromatin accessibility reveals disease mechanisms.** HCASMCs are primary cells that may not accurately mimic the native disease environment to prioritize CAD-associated mechanisms. Thus, we measured open chromatin via ATAC-seq on dissected medial layers from normal and atherosclerotic (athero) coronary arteries from human heart transplant donors and recipients, respectively. We observed a correlation ($r = 0.70$) between ATAC-seq tag densities from serum-free HCASMC and normal artery tissue (Fig. 3a), suggesting distinct yet similar epigenomes in native tissues versus cultured cells. We then performed a principle component analysis (PCA) to compare *ex vivo* open chromatin data sets with those from cultured

HCASMCs. Principal component 3 separated normal versus atherosclerotic tissues and stimulated cells by expected differentiation status (more to less differentiated, bottom to top), with HCASMC plus PDGF-BB, PDGF-DD or serum, and athero tissue in top half, and HCASMC serum-free, plus TGF-β, and normal tissue in bottom half (Fig. 3b). Principal component 2 separated the HCASMC donor lots, consistent with hierarchical clustering results (Fig. 1e). Given the cell- and tissue-specific regulatory contributions to CAD[36], we also performed PCA of HCASMC and *ex vivo* coronary artery open chromatin data with ENCODE DHS data, in which serum-treated HCASMCs and athero coronary tissue clustered with more de-differentiated cells

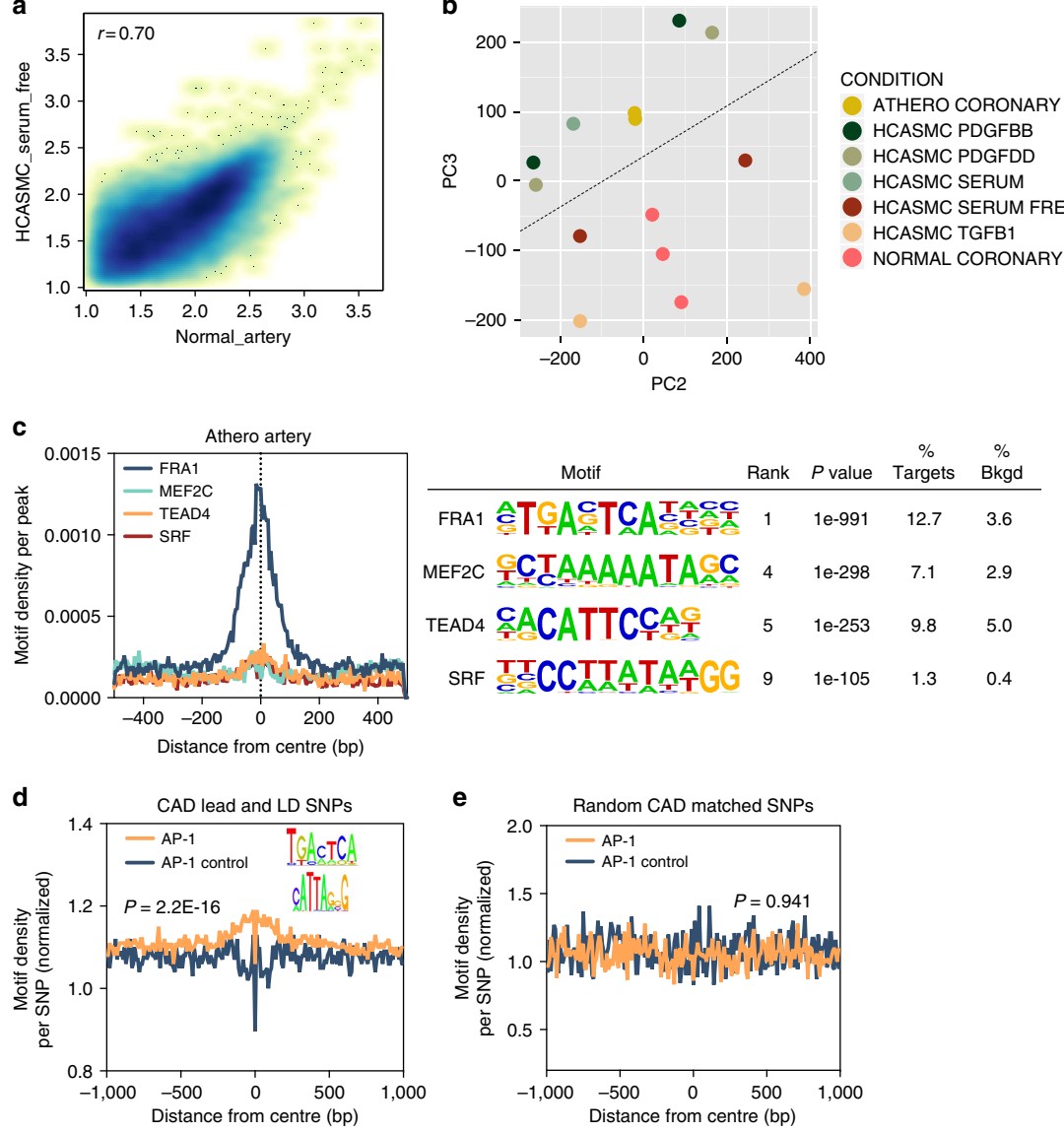

**Figure 3 | *Ex vivo* chromatin accessibility reveals upstream disease mechanisms. (a)** Scatter plot showing correlation of ATAC-seq tag intensities from normal coronary artery tissue and quiescent HCASMCs in serum-free conditions ($r^2 = 0.70$). Similar results were observed from $n = 3$ biological replicates. **(b)** Principal component analysis of ATAC-seq open chromatin peaks from normal and atherosclerotic coronary artery tissues and HCASMCs cultured under various conditions. Principal component 1 was excluded as it depicted a batch effect between experiments. Principal component 2 (PC2; *x* axis) represents the effect of the cell line used for treatment with various factors. Principal component 3 (PC3; *y* axis) represents the effect of the treatment with various factors and also partially separates normal and athero tissues (denoted by dashed line). **(c)** Histogram showing *de novo* motif distribution in *ex vivo* open chromatin peaks and resulting enrichment analysis in *ex vivo* open chromatin regions from atherosclerotic coronary artery tissues. Data represent $n = 2$ biological replicates. **(d–e)** Histogram density plot showing CAD lead and LD SNPs enriched in matrices for the classic AP-1 motif, TGANTCA, versus randomized AP-1 matrices (AP-1 control). Randomized CAD-matched SNPs centred on AP-1 or AP-1-randomized control matrices. Count densities were globally normalized by the total number of counts for both SNP and AP-1 motif regions in the 1-kb window.

(iPSC, stem cells) compared with normal coronary tissue (Supplementary Fig. 8). It is worth noting that these complex and limited human tissue samples present a challenge in resolving distinct clusters from a standard PCA. As demonstrated previously (Fig. 1), we identified the AP-1 family as the most significantly enriched motif (Fig. 3c; Supplementary Fig. 9) in *ex vivo* open chromatin peaks, followed by motifs for relevant factors MEF2C, TEAD4 and serum-response factor. To further investigate the relevance of AP-1 motifs in CAD, we correlated the positions of candidate CAD SNPs with classic AP-1 motifs across the genome and observed a peak of enrichment (Fisher's exact test $P = 2.2E - 16$), which was reversed using randomized AP-1 matrices (Fig. 3d). As a control, randomized CAD-matched SNPs displayed no gain or loss of enrichment with either AP-1 or control matrices (Fisher's exact test $P = 0.941$; Fig. 3e). Together, these data demonstrate that chromatin accessibility profiles of stimulated HCASMCs resemble those of diseased atherosclerotic tissue, and that AP-1-binding sites may explain a CAD-relevant *cis*-acting mechanism for the re-programming of these cells.

**AP-1- and TCF21-dependent regulatory mechanisms at CAD loci.** To disentangle specific CAD associations, we intersected the combined HCASMC-based ATAC, H3K27ac and TCF21/AP-1 ChIP-seq data with the CARDIoGRAMplusC4D variants, resulting in 87 candidate variants (30 variants *ex vivo*; Supplementary Data 7). We then annotated these variants for chromatin state and enhancer activity using ENCODE[37] and Roadmap Epigenomics data collected from >100 cells and tissues[38] (Supplementary Data 8), and utilized position weight matrices (PWMs) from JASPAR (http://jaspar.genereg.net) to annotate candidate variants predicted to alter TFBS motifs (http://www.broadinstitute.org/mammals/haploreg/haploreg.php; Supplementary Data 9). After filtering for variants with RegulomeDb (http://www.regulomedb.org/) functional scores ≤4 and evidence of protein binding *in vivo* (≥1 *trans*-acting factor), we obtained 64 (out of 87) variants in HCASMCs and 26 (out of 30) variants *ex vivo*. Interestingly, one of the lead candidate variants, rs17293632, located within an intergenic region of the *SMAD3* gene was linked to a new association for CAD, rs56062135 ($r^2 = 1.0$; $D' = 1.0$; Europeans) at 15q22.33, (additive model $P = 5.72E - 09$)[5] (Fig. 4a), while also being previously associated with chronic inflammatory bowel disease[39] and Crohn's disease[40] (Fig. 4a,b). *SMAD3* was previously associated with CAD through an independent SNP[41]; however, the variant described here, rs17293632, resides within an open chromatin region *ex vivo*, and preferentially in TGF-β- and PDGF-BB-treated HCASMCs (Fig. 4c,d), peaks for JUN and JUND binding, and H3K27ac marked active enhancer region (Fig. 4c). The major risk C allele was more associated with open chromatin (Fig. 4d) and further inspection revealed this allele to reside in a canonical AP-1 motif, TGACT[C > T]A, which is effectively destroyed by the minor protective T allele (Fig. 4c). Intriguingly, this is analogous to the AP-1-dependent mechanisms elicited by the *TCF21* risk alleles[10]. This variant, rs17293632, is also proximal to a muscle CAT motif for TEAD (involved in TGF-β signalling and SMC differentiation[42]) and upstream of a TCF21 motif, which we have shown to reside near AP-1 sites[35]. Using the *SMAD3* locus as a prototypical AP-1-dependent mechanism, we then performed allele-specific ChIP analyses for AP-1 factors, TCF21 and H3K27ac in HCASMCs from individuals who were heterozygous for rs17293632 ($n = 5$). These studies demonstrated preferential TF binding to the risk C allele, with non-allelic enrichment of H3K27ac (Fig. 4e). We also performed analyses for other top prioritized loci, such as rs1537373 at 9p21.3 (*CDKN2B/CDKN2B-AS1*), which overlapped

open chromatin, H3K27ac, AP-1 and TCF21 peaks (Fig. 5a). The risk T allele was associated with open chromatin (Fig. 5b), and TF and H3K27ac enrichment (Fig. 5c), despite being located outside but proximal to canonical motifs for AP-1 and TCF21. Finally, we investigated the mechanism for another variant, rs2019090, linked to the lead SNP, rs974819 ($r^2 = 1.0$; $D' = 1.0$; Europeans) at 11q22.3 (ref. 43) and located >150-kb downstream of the nearest coding gene, platelet-derived growth factor D (*PDGFD*). This variant resides in a modest open chromatin peak *ex vivo* and in stimulated open chromatin in HCASMCs, with the major risk T allele enriched in TF and H3K27ac peaks (Supplementary Figs 10 and 11). Similar observations were made through mapping of candidate variants at *LMOD1* and *IL6R* loci (Supplementary Figs 12 and 13). These findings demonstrate how integrative mapping approaches may point towards causal relationships that should be validated in functional studies *in vivo*.

**Allele-specific and total gene expression at CAD loci.** Mapping *cis*-regulatory mechanisms by detecting changes in binding to native chromatin implies effects on transcription. However, this assumption should be evaluated empirically. Thus, we identified seven top candidate causal variants to perform enhancer trap assays to measure allele-specific transcription, using regions of open chromatin flanking the candidate SNPs. The effects on transcription were modest, although significant allele-specific changes were observed at four loci, including rs17293632 (*SMAD3*), rs2019090 (*PDGFD*), rs7549250 (*IL6R*) and rs34091558 (*LMOD1*) (Fig. 6a). The greater effect size of rs17293632 (C > T) at *SMAD3* is likely attributed to its direct AP-1 motif disruption. To validate these effects at rs17293632, we performed gain and loss of function for AP-1 members, including JUN, JUNB and JUND. As expected, overexpression of JUN and JUNB further *trans*-activated the risk C allele reporter, consistent with the effects observed on the consensus AP-1 reporter (Fig. 6b; Supplementary Fig. 14). In contrast, short interfering RNA (siRNA)-mediated knockdown of AP-1 (predominantly JUN and JUNB) led to a reduction of the risk C allele compared with the protective T-allele reporter (Fig. 6c; Supplementary Fig. 14). Importantly, these AP-1 perturbations (particularly JUN) correlated with changes in *SMAD3* in HCASMCs (Supplementary Fig. 14).

We also observed an eQTL dosage effect at rs17293632 with the risk C allele correlated with greater endogenous *SMAD3* gene expression levels in HCASMC ($n = 64$, $P < 0.0001$; Fig. 6d). Similar results were observed for both *CDKN2B* and *PDGFD* candidate variants, rs1537373 and rs2019090, respectively (Supplementary Fig. 15). To further evaluate *cis*-effects on endogenous *SMAD3* levels, we performed AEI of SMAD3. Given the weak linkage of a proxy transcript SNP to detect allele-specific expression (ASE), we measured *SMAD3* pre-mRNA levels using individuals heterozygous at rs17293632 itself. This approach is supported by genome-wide methods correlating intronic reads to changes in transcription[44]. In all 23 heterozygous individuals, an allelic imbalance was observed for the C allele ($P = 8.72E - 13$) (Fig. 6e). Similarly, we detected significant allelic imbalance for other candidate loci using linked transcript SNPs at *PDGFD*, *LMOD1*, *CDKN2B* and *IL6R* (Supplementary Figs 16 and 17; Supplementary Data 16). As a positive control, allelic imbalance was also detected for the lead SNP at *TCF21*, rs12190287, as previously reported[45] (Supplementary Fig. 17). While the 3′-untranslated region variant at *CCDC97/TGFB1*, rs2241718, was predicted to affect binding, this variant might not alter endogenous *CCDC97* levels (Supplementary Fig. 16), but rather serve as an enhancer for neighbouring *TGFB1* in HCASMCs.

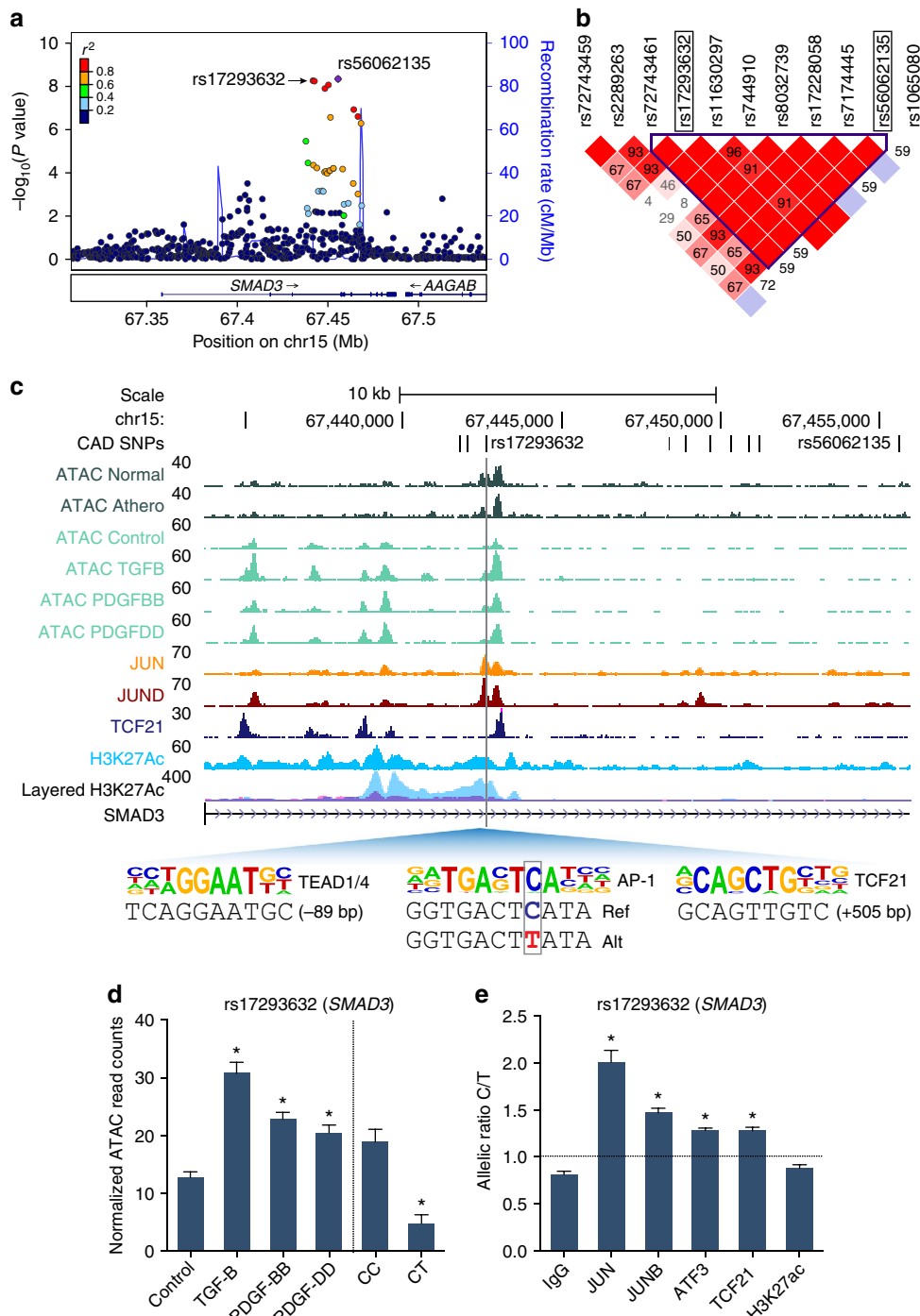

**Figure 4 | Functional mapping of *SMAD3* locus identifies AP-1-dependent regulatory mechanism. (a)** LocusZoom plot showing results of CARDIoGRAMplusC4D 1000G fine-mapping at *SMAD3* locus at chromosome 15q24.1 ($n = 60,801$ cases; $n = 123,504$ controls)[5]. Circles represent SNPs associated using an additive or recessive model, and colour-coded for LD ($r^2$) with the lead SNP, rs56062135 (purple diamond), based on 1000G phase 1 v3 training data set. **(b)** LD plot generated from 1000G phase 1 augmented haplotypes in Europeans for *SMAD3* locus, showing linked lead SNP, rs56062135, with candidate regulatory SNP, rs17293632. Colour-coded for LD based on $D'$ values, shown as in boxes. **(c)** UCSC browser screenshot at *SMAD3* locus, showing overlap of candidate SNP rs17293632 with ATAC-seq open chromatin tracks in coronary tissue ($n = 3$ biological replicates per condition) and HCASMCs treated under various conditions ($n = 2$ biological replicates per condition), TF-binding ChIP-seq tracks for TCF21, JUN and JUND, and active enhancer histone modification H3K27ac ChIP-seq ($n = 4$ biological replicates), as well as ENCODE layered H3K27ac for HUVEC (blue) and NHLF cells (purple). Inset, motifs in open chromatin regions showing alignment to reference sequence and position relative to rs17293632. Genomic coordinates refer to hg19 assembly. **(d)** Normalized ATAC-seq read counts for HCASMCs treated under various conditions and by genotype at rs17293632. Values represent mean ± s.e.m. ($n = 2$ biological replicates for stimulations and $n = 5$ biological replicates for different genotypes). **(e)** Allele-specific ChIP (haploChIP) for AP-1 proteins (JUN, JUNB and ATF3), TCF21 and H3K27ac in HCASMCs heterozygous at rs17293632. Values represent mean ± s.e.m. of triplicates from a representative experiment ($n = 5$ biological replicates). *$P < 0.01$ versus control, IgG or between two genotypes using an unpaired two-tailed *t*-test with Welch's correction for unequal variances.

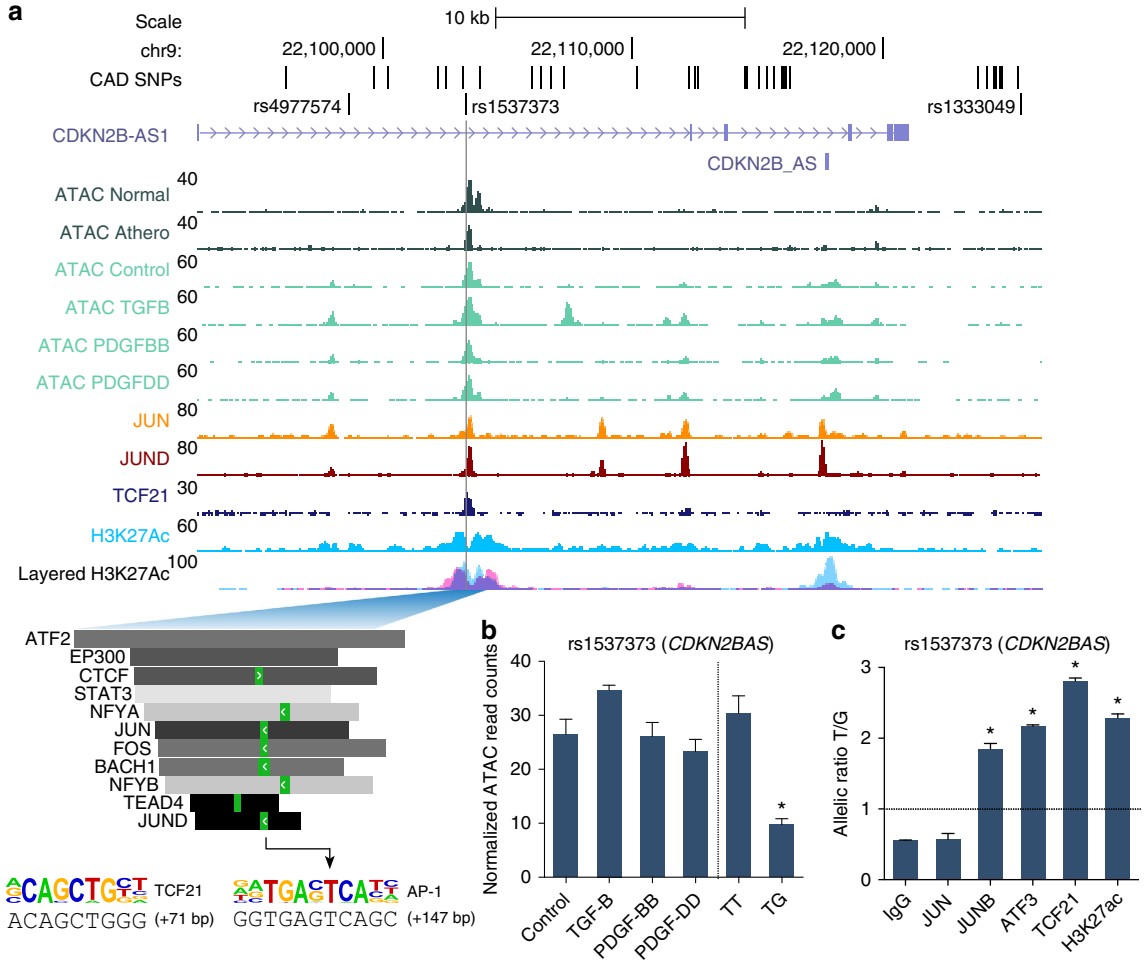

**Figure 5 | Functional mapping of 9p21.3 locus identifies AP-1- and TCF21-dependent regulatory mechanisms. (a)** UCSC browser screenshot at 9p21.3/*CDKN2B-AS* locus, showing overlap of candidate SNP rs1537373 with ATAC-seq open chromatin tracks in coronary tissue (*n* = 3 biological replicates per condition) and HCASMCs treated under various conditions (*n* = 2 biological replicates per condition), TF-binding ChIP-seq tracks for TCF21, JUN and JUND, and active enhancer histone modification H3K27ac ChIP-seq (*n* = 4 biological replicates), as well as ENCODE layered H3K27ac for HUVEC (blue) and NHLF cells (purple). Inset, motifs generated from HOMER in open chromatin regions showing alignment to reference sequence and position relative to rs1537373 SNP. Genomic coordinates refer to hg19 assembly. **(b)** Normalized ATAC-seq read counts for HCASMCs treated under various conditions and by genotype at rs1537373. Values represent mean ± s.e.m. (*n* = 2 biological replicates for stimulations and *n* = 5 biological replicates for different genotypes). **(c)** Allele-specific ChIP (haploChIP) for AP-1 proteins (JUN, JUNB and ATF3), TCF21 and H3K27ac in HCASMCs heterozygous at rs1537373. Values represent mean ± s.e.m. of triplicates from a representative experiment (*n* = 5 biological replicates). *$P < 0.01$ versus control, IgG or between two genotypes using an unpaired two-tailed *t*-test with Welch's correction for unequal variances.

Along these lines, it is predicted that the endogenous *trans*-acting factors upstream of these *cis*-regulatory elements must also be context dependent to elicit functional effects in HCASMCs. Thus, we performed a RNA-seq-based transcriptomic analysis of HCASMCs treated with TGF-β or PDGF-BB ligands to identify differentially expressed genes. Interestingly, using DESeq analysis of PDGF-BB treatment versus serum-free control HCASMCs, the most significantly altered genes were among the immediate early response genes: AP-1 members, *FOSB*, *FOS*, *JUN*, *JUNB* and *ATF3*, as well as *EGR3* (early growth response 3), and *NR4A1/2* (nuclear receptor subfamily 4 group A member 1/2; Fig. 6f; Supplementary Fig. 18; and Supplementary Data 10). These results strongly implicate immediate early response genes such as AP-1 factors as key *trans*-regulators of growth factor-dependent changes in chromatin accessibility, binding and expression at CAD loci in a critical vascular cell type.

**Global *cis*-eQTLs in external databases validate CAD variants.** To validate the endogenous function of our prioritized variants,

we performed a query in global *cis*-eQTL databases, including the Genotype-Tissue Expression (GTEx) project (Supplementary Data 11). All but one candidate variant (rs1537373) was detected as an eQTL in various tissues, with the lead candidate at SMAD3, rs17293632 strongly associated with SMAD3 levels in thyroid tissue ($P = 1.94E - 13$; Fig. 7b). This was not surprising given the homogenous cellular composition of thyroid tissue. Notably, IL6R variant (rs7549250) was one of the most significant eQTLs in whole blood (Fig. 7c), and the BMP1 candidate variant (rs73551707) was highly significant in aortic artery tissue (Fig. 7d). Given the limited sample number and disease phenotypes in these public databases, we then investigated our candidate SNPs in human atherosclerotic aortic tissues (*n* = 513) and internal mammary artery tissues (*n* = 528) collected during coronary artery bypass graft procedures from the STARNET (Stockholm-Tartu Atherosclerosis Reverse Network Engineering Task) database[23,46]. First, we searched for the most significant *cis*-eQTLs within a 2-Mb window surrounding each candidate SNP, and then performed a conditional analysis using the

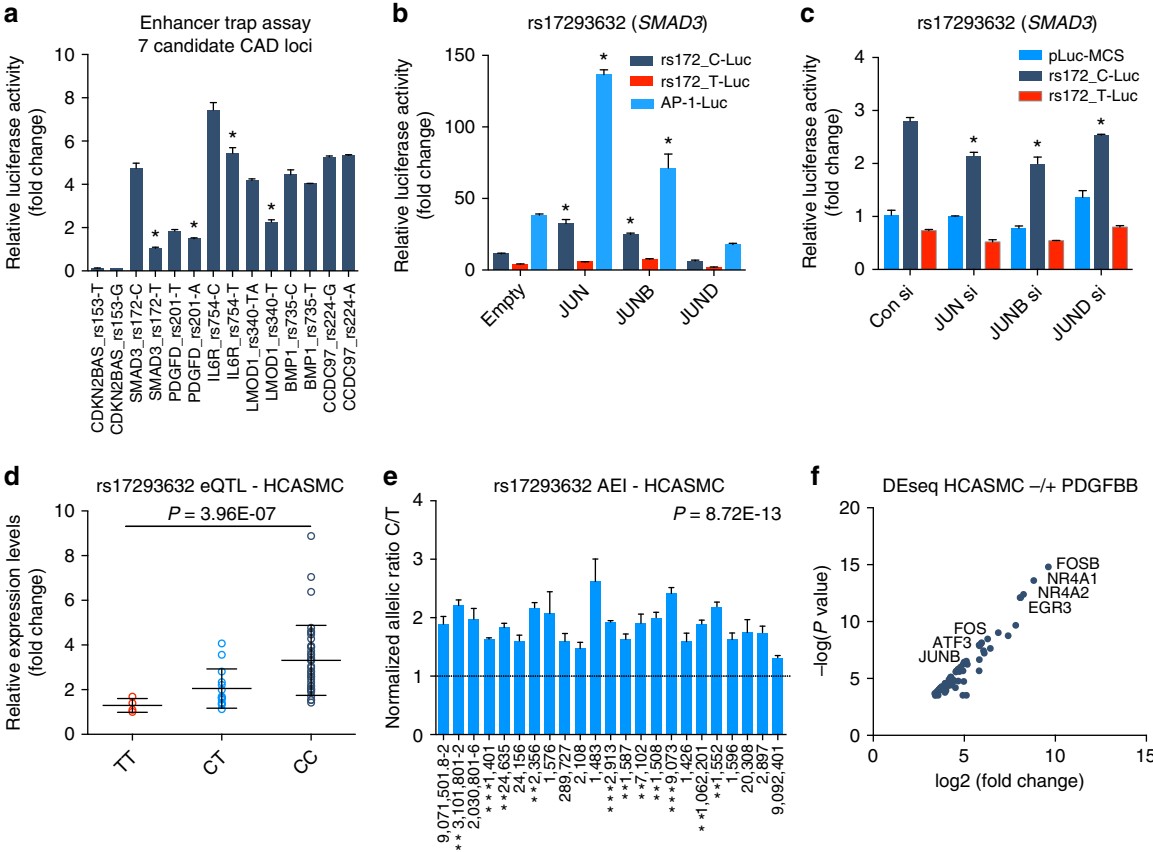

**Figure 6 | Allele-specific transcription and expression imbalance at CAD loci.** (**a**) Results of enhancer trap assay for seven candidate SNPs cloned into minimal promoter-driven luciferase reporter vector pLuc-MCS, and transfected in A7r5 smooth muscle cell line. (**b**) Results of enhancer trap assay for rs17293632-C/T (*SMAD3*) reporters (rs172_C-Luc and rs172_T-Luc) versus consensus AP-1-Luc (3 ×) reporter with co-transfection of expression constructs for JUN, JUNB or JUND in A7r5 SMC. (**c**) Results of enhancer trap assay for rs17293632 reporters with siRNA-mediated knockdown of JUN, JUNB or JUND in HCASMCs. (**a**–**c**). Values represent mean ± s.e.m. of triplicates for a representative experiment (*n* = 4 biological replicates), expressed as fold change relative to pLuc-MCS. *P < 0.01 between allele-specific reporters using a two-tailed Student's *t*-test. (**d**) *SMAD3* gene expression levels in HCASMCs with respect to genotype at rs17293632, expressed as ΔΔCt values normalized to *GAPDH* levels (fold change). Values represent mean ± s.e.m. of triplicates (*n* = 64 independent donors/biological replicates). *P* values calculated using a Welch's unequal variances *t*-test. (**e**) Allelic expression imbalance for candidate regulatory SNP rs17293632 at *SMAD3* detected by TaqMan qPCR in HCASMC pre-mRNA from heterozygous individual donors (*n* = 23). Values represent mean ± s.e.m. of triplicates for cDNA ratio normalized to gDNA ratio in heterozygous individuals at rs17293632. *P* values shown represent comparison of AEI from all samples versus expected allelic ratio of 1.0 using a Welch's unequal variances *t*-test. **P < 0.001, ***P < 0.0001 for individual samples of allelic imbalance ratio versus expected allelic ratio of 1.0. (**f**) Scatter plot for the most significant differentially expressed genes from RNA-seq DEseq analysis of HCASMCs treated with serum free (control) or PDGF-BB for 1 h (*n* = 2 biological replicates per condition). –log10(*P* values) and log2(fold change) determined as described in the Methods section. Labels are shown for immediate early response genes.

most significant SNP or the lead GWAS SNP in this region (Supplementary Data 12). Strikingly, we found that the candidate SNP at the 9p21.3 locus, rs1537373, was a significant *cis*-eQTL for *CDKN2B* (*P* = 2.13E − 05 in aorta; *P* = 0.0035 in mammary artery) and was the second most significant SNP in this entire region in aorta (Fig. 8a,b). We also observed greater *CDKN2B* expression at rs1537373 to correlate with the T allele (Fig. 8c), which is consistent with expected direction based on chromatin accessibility, TF binding, H3K27ac enrichment (Fig. 5) and expression analysis in HCASMCs (Supplementary Figs 14a and 16a). Furthermore, we identified a highly significant *cis*-eQTL for the candidate SNP, rs2019090, at the *PDGFD* locus (*P* = 2.34E − 21 in aorta; 2.41E − 13 in mammary artery), which represented the most significant *cis*-eQTL in this region for *PDGFD* in aortic tissue (Fig. 8d,e). Greater *PDGFD* expression at rs2019090 also correlated with the risk T allele (Fig. 8f), consistent with the expected direction based on analyses in HCASMCs (Supplementary Figs 15b and 16c). While we did observe multiple *cis*-eQTL genes for some top candidate SNPs,

such as rs73551707, these were nominally significant compared with the nearest target gene, *BMP1* (*P* = 1.45E − 15 in aorta; 1.10E − 14 in mammary artery; Supplementary Fig. 19; Supplementary Data 12). Others appeared to be tissue selective, for example, rs34091558 (*LMOD1*) and rs7549250 (*IL6R*) (Supplementary Fig. 19; Supplementary Data 12), which implicate context-specific gene regulation through these variants. Together, these analyses support the overall validity of our approach to deduce causal regulatory variation in the appropriate disease context.

## Discussion

Herein, we apply an integrative approach to investigate causal regulatory variants associated with a complex disease using genomic, epigenomic and transcriptomic analyses with targeted experimental follow-up at seven candidate loci. By incorporating haplotype information for 5,240 lead and high-LD GWAS variants along with custom genome-wide sequencing analyses for chromatin accessibility, active enhancers, TF binding, as well

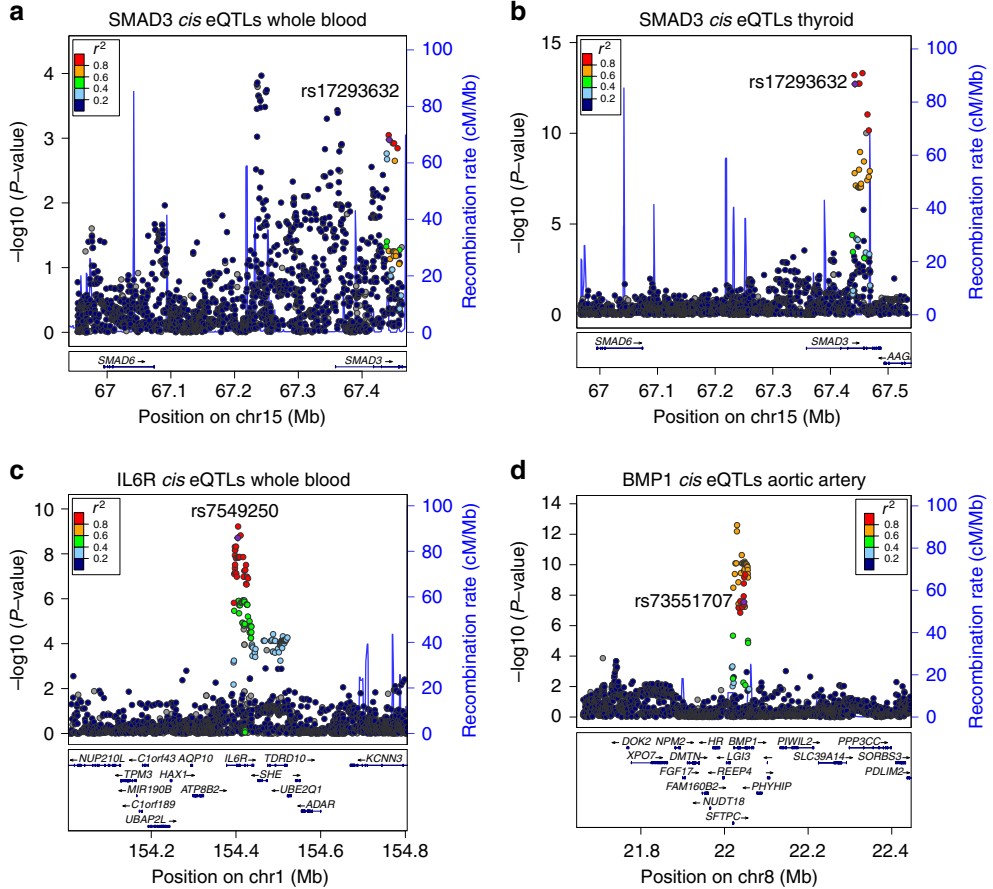

**Figure 7 | Validation of *SMAD3* and other candidate variants in GTEx eQTL database.** All SNPs associated with *SMAD3* gene expression in whole blood ($n = 338$) (**a**) and thyroid ($n = 278$) (**b**) tissue, which span the entire 500-kb locus, were extracted from the GTEx V6 database, and $-\log(P$ values) were plotted using LocusZoom. SNPs are colour-coded by $r^2$ value. Candidate SNP rs17293632 is shown in purple. Gene names are shown at genomic positions (hg19 assembly). (**c**) All SNPs associated with IL6R expression in whole blood ($n = 338$) are shown. Candidate SNP rs7549250 is shown in purple. (**d**) All SNPs associated with BMP1 expression in aortic artery tissue ($n = 197$) are shown. Candidate SNP rs73551707 is shown in purple.

as public annotations, we identify 64 candidate regulatory variants in stimulated HCASMCs and 26 candidates in coronary arteries *ex vivo* (Supplementary Fig. 1; Supplementary Data 4). We confirm the functionality of seven variants via allele-specific binding, enhancer traps and AEI (Supplementary Data 13 and 14). Using differential RNA-seq analysis, we identify *trans*-acting factors that are coupled to the *cis*-regulatory elements in growth factor-treated HCASMCs. Finally, genome-wide *cis*-eQTL mapping in arterial tissues from atherosclerotic diseased individuals further emphasizes that our HCASMC-based analyses may reflect the function of these variants in the appropriate disease environment (Supplementary Data 10).

Unique to our approach is the application of prior knowledge of CAD-associated TCF21 and AP-1 to prioritize other CAD regulatory variants. The *TCF21* locus was identified from the initial CARDIoGRAM meta-analysis of 14 GWAS in 22,233 cases and 64,762 controls of European descent[4], and was subsequently replicated in a meta-analysis in Han Chinese individuals[47]. While TCF21 is highly expressed in the developing epicardium and proepicardial organ to give rise to interstitial cells and coronary vasculature[48–50], it may also be critical for remodelling the adult artery during early atherogenesis[51]. TCF21 was selected given our recent findings showing enrichment of CAD loci in TCF21-binding sites in HCASMCs[35], which localize in the vicinity of AP-1 motifs[35]. Our findings that CAD loci are enriched for AP-1 motifs in open chromatin in both cells and tissues re-emphasizes the role of AP-1 in potentiating chromatin

accessibility[52]. We observed 58% of AP-1 (JUN and JUND) versus 81% of TCF21 peaks to overlap open chromatin peaks, suggesting that AP-1 factors are capable of priming compact chromatin for subsequent recruitment of inducible TFs. Whether AP-1 factors represent true 'pioneer factors' such as the FOXA or PU.1 factors[53] remains to be determined. Nonetheless, both TCF21 and AP-1 likely play important roles in remodelling the chromatin architecture to modulate SMC fate and differentiation.

One of the CAD-associated enhancers we identified at the *SMAD3* locus has been associated with IBD[39] and Crohn's disease[40], and was highlighted in a systematic scan of autoimmune disease loci[54]. The fact that the rs17293632 risk allele resides in a classical AP-1 motif parallels our previous observations for the *TCF21* risk alleles[10], and may support a 'multiple enhancer variant' model for common regulatory pathways[55]. More importantly, we note that open chromatin regions in HCASMCs were also highly enriched for diseases of chronic inflammatory origin, including UC, Crohn's disease and RA (Fig. 2e). While the hypothesis that inflammation plays a causal role in atherosclerosis is not novel, for example, through the actions of modified lipoprotein-activated macrophages[56,57], it is intriguing to speculate that SMCs are active participants in the early immune responses through their '*trans*-differentiation' into macrophage-like foam cells[58]. This hypothesis is strengthened by recent evidence that one of the encoded proteins at the top CAD-associated 9p21.3 locus, cyclin-dependent kinase inhibitor 2B (CDKN2B/p15), controls the clearance of SMC-derived apoptotic

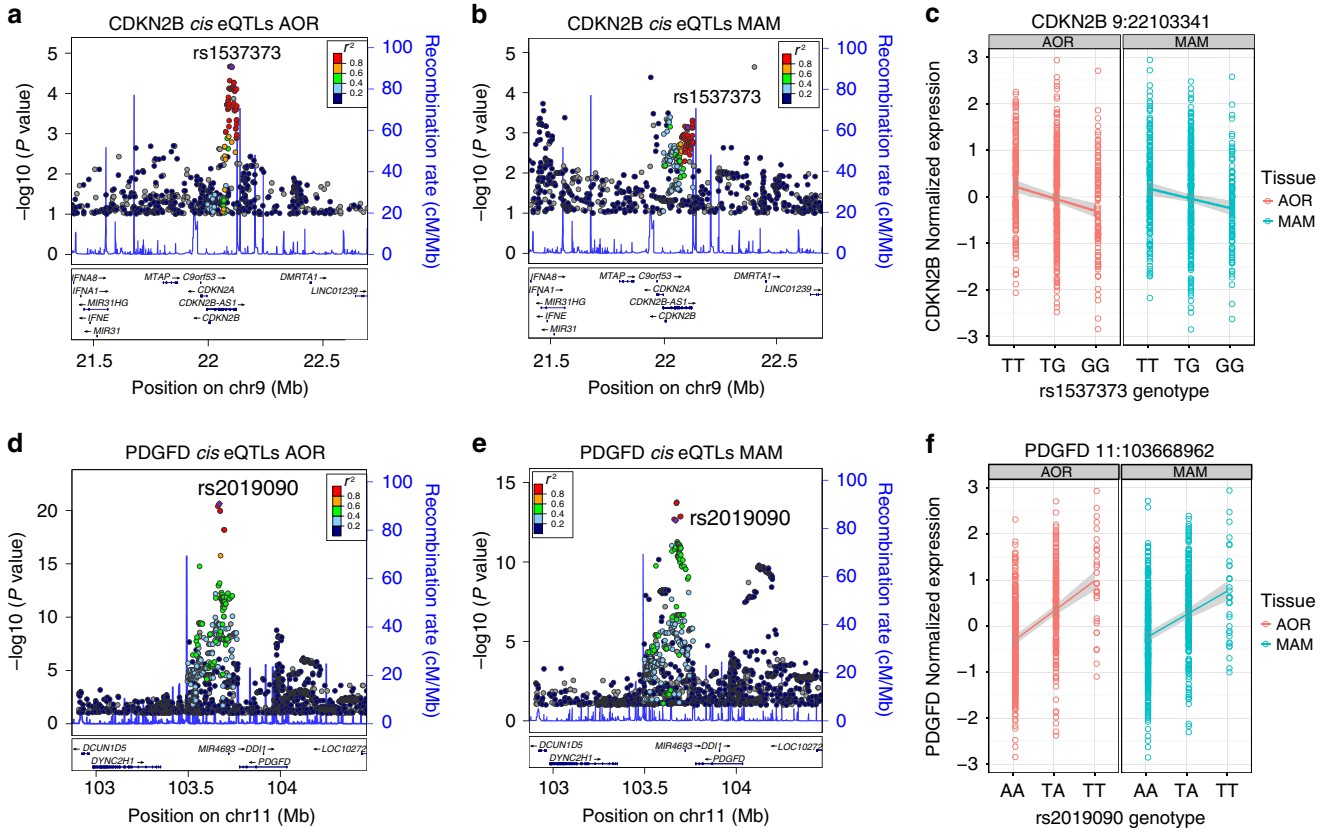

**Figure 8 | Validation of *CDKN2B* and *PDGFD* candidate variants in STARNET CAD eQTL data sets.** (**a,b**) Distribution of all *cis*-eQTLs called for *CDKN2B* from STARNET diseased human aorta (*n* = 513 independent donors/biological replicates) and mammary artery (*n* = 528 independent donors/biological replicates) tissues, respectively. Association plot includes variants colour-coded by $r^2$ value to represent LD with candidate SNP (shown in purple). Gene names are shown at genomic positions (hg19 assembly). (**c**) For each genotype, the normalized expression of *CDKN2B* in STARNET is shown as well as the linear regression line. (**d,e**) Distribution of all *cis*-eQTLs called for *PDGFD* in aortic and mammary artery tissue. In this case, the candidate SNP of interest was also the most significant eQTL for *PDGFD*. (**f**) For each genotype, the normalized expression of *PDGFD* is shown as well as the linear regression line. −log10(*P* values) and log2 normalized expression levels determined as described in the Methods section. AOR, aorta; MAM, mammary artery.

bodies and regulates inflammatory cytokine production during atherosclerosis[59]. We also investigated the mechanisms of a putative 9p21.3 enhancer for the antisense non-coding RNA *CDKN2B-AS1*. This variant, rs1537373, resides in a large haplotype block of linked variants including the highly replicated CAD SNPs, rs4977574 and rs1333049 (ref. 41), and while it does not directly alter a known TF binding motif, it localized to a region of accessible chromatin, TF binding and enhancer activity (Fig. 5), proximal to both TCF21 and AP-1 consensus motifs. Allele-specific TF and H3K27ac enrichment at rs1537373 implicate changes in the native chromatin structure, consistent with the observed *cis*-eQTL for *CDKN2B* (rather than *CDKN2B-AS1*) in aortic tissues (*P* = 2.18E − 05; Fig. 8; Supplementary Data 12). This variant was strongly associated with coronary artery calcification[60], and warrants follow-up study to unravel the complex pleiotropic and physical interactions at this locus.

Similarly, we investigated the causal mechanism at the 11q22.3 locus through a variant, rs2019090, which is linked ($r^2$ ∼ 0.97, Europeans/Asians) to the lead GWAS variant, rs974819, identified in individuals of European and South Asian ancestry[43]. This variant is located ∼360-kb downstream of the transcription start site (TSS) for *PDGFD*, resides in a moderately active enhancer region in HCASMCs, and alters binding and chromatin accessibility despite being outside consensus TF-binding motifs (Supplementary Fig. 10). Importantly, we identify this variant as the most significant *cis*-eQTL for *PDGFD* in diseased aortic

tissues (*P* = 2.34E − 21; Fig. 8; Supplementary Data 12). These data strongly implicate rs2019090 and *PDGFD* as the likely causal variant and gene at this locus. This also supports the notion that variants converging on weaker and distal enhancers may exert meaningful functional effects on endogenous genes. While both *CDKN2B* and *PDGFD* likely play critical roles in early-stage SMC disease processes, these effects may not be restricted to SMC. For instance, PDGFD is an important paracrine signal from neighbouring endothelium and adventitia and may mediate functions independent of the classic PDGF-BB ligand[61]. Recent data that PDGFRβ signalling accelerates local inflammation and hypercholesterolemia through the outer arterial layers (adventitia and media)[62], support a critical role of either SMC or pericyte activation. One of our candidate loci, *LMOD1* (Leiomodin 1), does appear to have a SMC-restricted expression pattern as a serum-response factor/myocardin target gene involved in actin filament assembly[63]. The candidate enhancer variant we identified for *LMOD1*, rs34091558, was a significant *cis*-eQTL in aorta but not mammary artery (Supplementary Fig. 13; Supplementary Data 12), suggesting differences in SMC content or differentiation state during disease. The *IL6R* candidate variant, rs7549250, was shown to be a mammary artery selective *cis*-eQTL (Supplementary Fig. 18; Supplementary Data 12), which may reflect the more systemic actions of this pro-inflammatory cytokine pathway. It is worth noting that this receptor is currently targeted for RA and juvenile idiopathic arthritis[64], and potentially for sub-clinical atherosclerosis[65].

In summary, these studies highlight the utility of combining multiple GWAS and genomic data sets to uncover the mechanisms of a complex disease such as CAD. We expect many of these epigenetic changes to involve activation of SMCs; however, defining cell specificity will require further investigation. SMCs are one of many vascular cell types linked to athero-sclerosis, including endothelial cells, monocytes, and macro-phages/foam cells, however non-vascular cell types such as hepatocytes may also be indirectly responsible for CAD pathology through lipid metabolism[4,5]. While these studies of a single vascular cell type have inherent limitations, future large-scale efforts to investigate the crosstalk between vascular and non-vascular cells may provide further mechanistic insight into disease causality. Efforts to directly modify these associated regulatory elements or genes in specific cell populations *in vivo* using CRISPR/Cas9 may provide insights into the direction of effect involving other disease processes[66]. Thus, by combining systematic predictive analyses[67,68] with exhaustive genome-wide functional mapping of candidate risk loci in disease-relevant tissues or individual cells[69], followed by genomic targeting in normal and disease states, such integrative approaches may finally unravel the causal regulatory mechanisms of disease for early-stage detection and/or therapeutic development.

## Methods

**Primary cell culture and reagents.** Primary HCASMCs derived from normal human donor hearts were purchased from three different manufacturers, Lonza, PromoCell and Cell Applications (all tested negative for mycoplasma contamina-tion). HCASMCs were maintained in growth-supplemented smooth muscle basal media (Lonza) according to the manufacturer's instructions. All experiments were performed on HCASMCs between passages 4 and 7. Rat aortic smooth muscle cells (A7r5) were purchased from ATCC (CRL-1444) and maintained in Dulbecco's modified Eagle media supplemented with 10% fetal bovine serum according to the manufacturer's instructions. Both HCASMCs and A7r5 (not listed in database of misidentified cell lines maintained by ICLA) were validated by immunostaining for alpha-smooth muscle actin using a mouse monoclonal antibody purchased from Sigma (A2547). Human recombinant PDGF-BB, PDGF-DD and TGF-β1 (containing bovine serum albumin carrier protein) were purchased from R&D Systems and were used at 20, 10 and 10 ng ml$^{-1}$, respectively. Antibodies used for ChIP-seq and ChIP–quantitative PCR (qPCR) were all pre-validated according to ChIP-seq guidelines and ENCODE best practices[70]. Purified rabbit polyclonal antibodies against human JUN (sc-1694 X), JUNB (sc-46 X) and JUND (sc-74 X) were purchased from Santa Cruz. Purified rabbit polyclonal antibody against H3K27ac (ab4729) was purchased from Abcam. Purified rabbit polyclonal antibody against human TCF21 (HPA013189) was purchased from Sigma.

**ATAC-seq analysis in HCASMCs.** ATAC-seq was performed with slight modifications to the published protocol. Briefly, HCASMCs (passages 5–6) were cultured in normal media until ~75% confluence. Approximately 5.0E4 fresh cells were collected by centrifugation at 500*g* and washed twice with cold 1× PBS. Nuclei-enriched fractions were extracted with cold lysis buffer containing 10 mM Tris–HCl, pH7.4, 10 mM NaCl, 3 mM MgCl$_2$ and 0.1% IGEPAL (octylphenox-ypolyethoxyethanol), and the pellets were resuspended in transposition reaction buffer containing Tn5 transposases (Illumina Nextera). Transposition reactions were incubated at 37 °C for 30 min, followed by DNA purification using the DNA Clean-up and Concentration kit (Zymo). Libraries were initially PCR amplified using Nextera barcodes and High Fidelity polymerase (NEB). The number of cycles was empirically determined from an aliquot of the PCR mix, by calculating the Ct value at 25–30% maximum Rn for each library preparation. The final amplified library was again purified using the Zymo DNA Clean-up and Concentration kit, and the DNA was evaluated by TBE gel electrophoresis and quantified using Bioanalyzer, nanodrop and quantitative PCR (KAPA Biosystems). Libraries were multiplexed and paired-end 50-bp sequencing was performed using an Illumina HiSeq 2500. Raw FASTQ files were evaluated using a modification of the FastQC pipeline to generate per base and per sequence-level summary statistics. Libraries that achieved consistent high-quality scores from this tool were accepted and paired-end reads were aligned to the human genome (hg19) using Bowtie2 with the –X 2000 maximum insert size parameter. Mitochondrial reads were not excluded as these represented <5–10% total reads. After adjusting the read start positions to account for Tn5 insertion bias using the preShift.pl script, peaks were called using Model-based Analysis for ChIP-Seq (MACS) with –P value 1.0E − 05 cutoff to reveal open chromatin peaks. Bigwig files were generated for University of Cali-fornia Santa Cruz (UCSC) visualization, and peaks were annotated using HOMER. To calculate the differential chromatin accessibility between treatments,

normalized read counts generated from HOMER were combined into a matrix and a generalized linear model likelihood ratio test was used to compute *P* values with the edgeR R/bioconductor package (https://bioconductor.org/packages/release/bioc/html/edgeR.html) using default parameters. These results are summarized in Supplementary Data 17. Alternatively, normalized read counts at top candidate loci were subjected to unpaired two-tailed *t*-test with Welch's correction for unequal variances.

**Ex vivo ATAC-seq analysis in coronary artery tissues.** Coronary artery tissues (left and right main arteries) were dissected from freshly explanted hearts from consenting donors (under approved Institutional Review Board protocol at Stanford University) at the time of heart transplantation. Normal arteries were obtained from rejected donor hearts, and atherosclerotic arteries were obtained from ischaemic diseased hearts from transplant recipients. Adventitial layers were carefully removed from coronary artery segments before snap-freezing medial layers in liquid nitrogen and storage at − 80 °C. Medial tissues from either normal or atherosclerotic arteries were pulverized using a liquid nitrogen chilled steel homogenizer. An amount of 10–20 mg of pulverized fresh-frozen tissue was suspended in 1 ml 1× PBS and centrifuged at 2,000*g* for 3 min at 4 °C. The pellet was resuspended in 1 ml of a lysis buffer containing 50 mM HEPES, pH 7.5, 140 mM NaCl, 1 mM EDTA, 10% glycerol, 0.5% IGEPAL CA-630 and 0.25% Triton X-100, and supplemented with an EDTA-free complete protease inhibitor (Roche). This suspension was incubated with rocking at 4 °C for 10 min and then homogenized via 15 loose strokes in a glass dounce. The homogenate was centrifuged at 2,000*g* for 5 min at 4 °C, and the resultant pellet was resuspended in 1 ml 1× PBS. Cellular debris was filtered out with a 40-μm cell strainer, and intact nuclei were quantified. A total of 50,000 nuclei were subjected to Tn5-mediated transposition for 1 h and tagmented DNA was amplified and purified as previously reported. Libraries were subjected to an additional electrophoresis step on an E-Gel EX 2% agarose gel (Invitrogen), and gel fragments correlating to 100–1,000 bp were excised and purified. The quality of the library preparation was determined by evaluation of the electropherogram traces from an Agilent 2100 Bioanalyzer DNA High Sensitivity, and libraries demonstrating appropriate nucleosomal enrichment were multiplexed and subjected to a lane of Illumina HiSeq 2500 sequencing. Sequences were analysed as described above.

**ChIP-seq analysis.** HCASMCs were cultured in normal media containing serum and fixed in 1% formaldehyde to crosslink chromatin, followed by quenching with glycine. 2.0E07 cells were collected, and nuclear lysates were prepared using dounce homogenization (20 strokes) in cold hypotonic buffer, followed by lysis in 1× RIPA buffer (Millipore). Chromatin nuclear lysates were then sheared to fragments of 100–500 bp using a Bioruptor Pico sonicator (Diagenode) according to the manufacturer. An amount of 5 μg of anti-rabbit IgG, TCF21 (Sigma; HPA013189), JUN (Santa Cruz; sc-1694), JUND (Santa Cruz; sc-74) or H3K27ac antibody (Abcam; ab4729) was added to sheared chromatin to immunoprecipitate TF–DNA complexes overnight at 4 °C. Following capture of the antibody–TF–DNA complexes to Protein G beads, the complexes were washed with ice-cold 1× RIPA and 1× PBS and eluted twice in 1× TE containing 1% SDS for 10 min each at 65 °C. Protein–DNA crosslinks were reversed overnight at 65 °C and ChIP DNA was recovered using Qiagen PCR Purification kits. To generate the ChIP library, Illumina TruSeq adapters were ligated to the ChIP DNA, followed by PCR amplification and gel electrophoresis on a 2% agarose gel. ChIP DNA library fragments ~300 bp were selected for PCR amplification. PCR reactions were performed under the following conditions 98 °C 30 s; (98 °C 10 s; 65 °C 30 s; 72 °C 30 s) × 14 cycles; 72 °C 5 min. The ChIP DNA library concentrations were quantitated by Qubit fluorometric and bioanalyzer analyses. Libraries were sequenced on an Illumina HiSeq 2500 (2 × 101) to obtain ~45–50 million reads per barcoded sample. Resulting fastq files were aligned to human genome hg19 using bowtie2 to generate bam files and peaks were called using MACS 1.4 with treatment sample as TCF21, JUN, JUND or H3K27ac, and control sample as IgG using the regional and local lambda model (TFs) or regional lambda (H3K27ac), bandwidth of 300 and a *P* value threshold of 1.0E − 05.

**TFBS analysis.** To identify DNA-binding sites enriched in ATAC-seq open chromatin peaks, AP-1, TCF21 or H3K27ac ChIP-seq peaks, the HOMER (http://homer.salk.edu/homer/ngs/) findMotifsGenome.pl script was employed to search for known TRANSFAC motifs and to generate *de novo* motifs[59]. All software parameters were set to default values, with the addition of the '-size given' command to define the width of each peak from the data rather than a constant value. Motifs discovered by HOMER were validated with MEME-ChIP[60] with a maximum motif length of 10. The motifs identified by MEME-ChIP were further compared with the binding motifs of known TFs. Density plots of the top *de novo* motifs were generated as follows. ATAC open chromatin or ChIP-binding summits were defined using MACS with default parameters for the collection of all identified binding regions. Motif distribution plots were generated using HOMER annotatePeaks.pl script centred on the respective motifs using the location of the summits within each peak file. TRANSFAC matrices for the top HOMER known-motif outputs were used to scan open chromatin and binding summit locations in

the human GRCh37/hg19 genome. Scanning for motifs was performed using annotatePeaks.pl with the following parameters: hg19-size 2000-hist 20.

**Super-enhancer analysis.** Super enhancers were defined using HOMER findPeaks tool with parameters findPeaks < tag directory> -i <input tag directory> -style super -o auto. In total, 24,727 H3K27ac peaks were used to define in total 653 super enhancers and a super-enhancer stitching window of 12,500 bp. FDR rate threshold was 0.001 and FDR effective Poisson threshold was 2.69E − 06.

**Footprinting analysis.** To detect genomic footprints of physical protein–DNA binding within regions of accessible chromatin, we employed the Wellington algorithm[61]. The wellington_footprints.py script was used with the updated –A parameter to account for the Tn5 integration bias in the ATAC-seq data, and footprints spanning all open chromatin regions were called at an FDR of 0.01. We then centred the generated ATAC footprints on CTCF motifs using the HOMER annotatePeaks.pl script, and utilized the dnase_average_profile.py and dnase_to_javatreeview.py scripts with the –A parameter and the respective motif centred intervals to generate histograms and heatmaps to plot the distribution of Tn5 integration.

***Cis*-regulatory functional enrichment and network analysis.** To annotate the function of HCASMC-stimulated ATAC-seq open chromatin sites, we utilized the GREAT (Genomic Regions Enrichment of Annotations Tool) algorithm. Test genomic regions were uploaded to the GREAT webserver and the entire hg19 human genome was used as a background. Genes were associated with the test genomic regions using the 'basal plus extension' parameters, which defines gene regulatory domains within a proximal region of 5-kb upstream, 1-kb, downstream and a distal region extending up to 1,000 kb from the TSS. Annotations from various biological, molecular and disease-based ontologies were used to associate genomic regions to functional annotations. Also, we used the aligned reads from HCASMC-stimulated ATAC-seq with the HOMER annotatePeaks.pl script to identify normalized read counts at TSS of nearest gene using the 'tss' and -size -500,100 parameters. These regions were intersected with CAD loci to perform k-means hierarchical clustering using a matrix of log normalized read counts across treatments, and heatmaps were generated using the ggplot2 R package. The CAD-associated gene list was also used with the Genes2FANs (functional association networks) algorithm to identify potential interactions from known functional association databases (including TFs, TRANSFAC and protein–protein interactions). The maximum path length to detect intermediate nodes connected to the seed nodes and z-score significance cutoff were set at 2.0.

**GWAS enrichment analysis in accessible chromatin regions.** GWAS SNP positions were downloaded from the NHGRI-EBI GWAS catalogue. ATAC-seq open chromatin regions were centred and GWAS SNPs were counted in a window of 100 bp for ± 1 kb surrounding centred ATAC-seq regions using the Feature correlation tool from the ChIP-Cor module (part of ChIP-Seq Analysis Server of the Swiss Institute of Bioinformatics (http://ccg.vital-it.ch/chipseq/chip_cor.php)). Counts were normalized to the total number of reference counts. The P values for enrichment of GWAS SNPs in ATAC-seq open chromatin regions were calculated as described previously[18]. The P values were computed using binomial cumulative distribution function $b(x;n,p)$ in R (dbinom function). We set the parameter $n$ equal to the total number of GWAS SNPs in a particular GWAS phenotype. Parameter $x$ was set to the number of GWAS SNPs for a given GWAS phenotype that overlap ATAC-seq regions and parameter $p$ was set to the fraction of the uniquely mappable human hg19 genome (2,630,301,437 bp) that is localized in the ATAC-seq open chromatin regions and contains assessed GWAS phenotype SNPs. Calculated binomial P value equals the probability of having $x$ or more of the n test genomic regions in the open chromatin domain given that the probability of that occurring for a single GWAS genomic region is $p$.

Custom R script was generated to intersect data sets and calculate Fisher's exact P values. Peak intervals from each data set were selected to contain those that have at least 1 bp overlap with a combined set of ENCODE DHS intervals from 125 ENCODE cell lines. ATAC-Seq, and JUN, TCF21, H3K27ac ChIP-Seq intervals were tested either separately or intersected (ATAC-JUN, ATAC-TCF, ATAC-K27, ATAC-K27-JUN and ATAC-K27-TCF). For each analysis, we determined overlaps with CAD SNPs, IBD SNPs, UC SNPs or a whole GWAS catalogue minus CAD SNPs, as a control. Overlaps between datasets, as well as sites present in one or the other tested data set, were determined using the ENCODE DHSs as a background. Finally, we determined DHSs devoid of each of the tested data set intervals. One-sided Fisher's exact test was used to determine whether the observed peak overlap between two tested data set intervals was statistically greater when compared against a background of all DHS peaks. Contingency matrices were made in R and Fisher's exact test was performed using fisher.test R command.

**PCA using ENCODE DHS data sets.** Processed DNase-seq data for 125 cell types generated by the ENCODE Analysis Working Group with the uniform processing pipeline (Uniform DNaseI HS track for Human Genome Build 37 (hg19)) were collected from the UCSC Golden Path FTP server. Each of 125 bedgraph files was

merged using the bedtools merge tool to contain only unique and non-overlapping intervals with a mean intensity for the overlapping intervals. Bedgraph files were combined to make a genome-wide matrix using bedtools unionbedg tool and were filtered for the regions smaller than 100 bp using a custom script, since these small regions mainly represent edges of the intervals that were not identically called in different DNase-seq files and would influence the downstream variance analysis. Furthermore, peaks on chrX and chrY were filtered out to eliminate the gender effect on the downstream PCA analysis of variance. To eliminate the influence of extreme outliers, we selected only data points in the range 100–2,000 of the absolute peak intensity. Data generated with the ATAC-seq method were processed using MACS (MACS v1.4.2) pipeline. ATAC-seq data were normalized to encompass the same range of data points as the DNase-seq data using the parameter equal to the ratio of the average peak intensity for each ATAC-seq data set and the average peak intensity for the ENCODE AoSMC cell type (as this represents the closest cell type to HCASMCs used for ATAC-seq). Each of the values from ATAC-seq data sets was divided with the calculated factor for that data set. Sex chromosomes were removed and intervals smaller than 100 bp were filtered out as described above. PCA was computed in R using the prcomp function, and PCA plots for principal components 2 and 3 were generated using the ggplot2 package and visualized with the wesanderson colour palette package for R available on github.

**SNP motif density plot calculation.** CAD and SNPs in LD were defined using the LD threshold of 0.8. Full-length AP-1 (JUN::FOS) motif (MA0099.1) was randomized using JASPAR (http://jaspar.genereg.net). The randomized model was created by permuting the columns so that the base composition of the randomized matrix remains the same as the JUN::FOS matrix to eliminate potential base composition bias. Whole-genome matrix scan was performed with PWMScan—genome-wide PWM scanner (http://ccg.vital-it.ch/pwmtools/pwmscan.php). PWM scan threshold was set to 0.001 to get approximately same number of sites as in AP-1 (randomized motif scan against hg19: 4114183 hits; AP-1 motif 10650673 hits). Density plots were calculated with the ChIP-Cor Analysis Module (http://ccg.vital-it.ch/chipseq/chip_cor.php) for AP-1 and AP-1-randomized matrix using the global normalization parameter and matching motifs in both DNA strands. Correlation was repeated with the randomly selected set of 5,265 SNPs from the set of SNPs in low LD with CAD SNPs (<0.5).

We also counted the total number of PWM sites for regions ± 100 bp from the SNPs (that is, where the enrichment is seen in the AP-1 matrix density profile and a gap is seen in AP-1-randomized matrix) and for 900–1,000 bp away (background regions) for the CAD GWAS and control SNPs, and obtained a contingency matrix (173,761, 63,176, 165,206 and 65,423, that is, AP-1-CAD SNPs, AP-1-randomized control matrix-CAD SNPs, AP-1-background regions and AP-1-randomized control matrix-in background regions) that gave us Fisher's exact P value of $P < 2.2E − 16$. Similar calculation for randomized matched SNP controls gave us non-significant P-values ($P = 0.9491$).

**Allele-specific ChIP–qPCR (HaploChIP).** Heterozygous genotypes at the candidate loci were determined using TaqMan SNP genotyping qPCR assays (Supplementary Data 15). SNP genotyping assays were further validated using PCR-based Sanger sequencing. Briefly, heterozygous HCASMC lines were cultured under normal conditions, growth factor stimulated, and chromatin crosslinked, sheared and immunoprecipitated as described above. Purified DNA was then amplified using TaqMan SNP genotyping assay probes against the candidate SNPs. Calibration of the SNP genotyping assay was determined by mixing 10 ng of HCASMC genomic DNA (gDNA), homozygous for each allele at the following ratios: 8:1, 4:1, 2:1 1:1, 1:2, 1:4 and 1:8. The Log2 ratio of the VIC/FAM (VIC-proprietary, Applied Biosystems; FAM – 6-carboxyfluorescein, Applied Biosystems) intensity at cycle 40 was then plotted against the Log ratio of the two alleles to generate a linear regression standard curve. The Log ratio of the intensity of the two alleles from complementary DNA (cDNA) samples was fitted to the standard curve. These values were then normalized to the ratio of gDNA for each allele to obtain the normalized allelic ratio. The Log2 ratio of VIC/FAM intensity at cycle 40 was then fitted to the standard curve and normalized to the gDNA ratio.

**Enhancer trap luciferase reporter assays.** Oligonucleotides containing the regulatory elements overlapping candidate CAD loci (156 nt for 1 × construct or 468 nt for 3 × construct) were synthesized for each allele and cloned into a shuttle vector pUC57 (GeneWiz). These fragments were then isolated by double restriction digest and subcloned into the multiple cloning site of the minimal promoter containing pLuc-MCS luciferase reporter vector (Agilent). All constructs were validated by Sanger sequencing. Empty vector (pLuc-MCS), 1 × and 3 × CAD loci and *Renilla* luciferase constructs were co-transfected, with or without pcDNA3-empty, pcDNA3-JUN, JUNB or JUND expression constructs into A7r5 or HCASMCs using Lipofectamine 3000 (Life Technologies) according to the manufacturer's instructions. For siRNA knockdown studies, 20 nM Silencer Select siRNA negative control or siRNA against JUN, JUNB or JUND (Life Technologies) was transfected in HCASMCs 12 h before reporter transfections using RNAiMAX (Life Technologies). Media was changed after 6 h, and dual-luciferase activity (Promega) was recorded after 24 h using a SpectraMax L luminometer (Molecular

Devices). Relative luciferase activity (firefly/*Renilla* luciferase ratio) is expressed as the fold change of the empty vector control (pLuc-MCS).

**AEI assays.** Total RNA was isolated from >60 primary HCASMCs from unrelated donor tissues, purchased from three different vendors (Lonza, Cell Applications and PromoCell) using the miRNeasy Mini kit (Qiagen) according to the manufacturer's instructions. Total cDNA was prepared using the High Output cDNA Synthesis kit (Applied Biosystems) according to the manufacturer's instructions. Total gDNA was prepared using the Blood and Tissue DNA Isolation kit (Qiagen). To identify appropriate markers for detecting allelic imbalance or ASE, linked coding or untranslated region SNPs were identified in haplotype blocks of genomic regions surrounding candidate regulatory SNPs using 1,000 Genomes Phase 1 or 3 variants (http://browser.1000genomes.org/) in Europeans (depending on the LD data availability for each SNP). $D'$ values of LD >0.90 were considered as the cutoff for linked variants; however, moderately linked variants with $D' > 0.75$ were used if necessary. In the case of SNP rs17293632, given the weak linkage between the proxy transcript SNP rs1065080, we used the intronic SNP rs17293632 itself to measure imbalance in the nascent SMAD3 pre-mRNA as a transcription surrogate[44]. Another criteria for detecting ASE with these markers was minor allele frequency (MAF) >0.15 in Europeans (as determined from 1000 Genomes data), since this is a limiting factor to obtain sufficient heterozygous individuals from the cohort of HCASMCs. Refer to Supplementary Data 16 for details on these proxy transcribed SNPs. TaqMan SNP genotyping probes were ordered for the identified ASE markers (Life Technologies) and calibrated in homozygous genotyped individual samples for each locus, as described above. SNP genotyping assays were further validated using PCR-based Sanger sequencing. ASE was then evaluated in heterozygous individual samples by determining the ratio of transcript levels for each allele using cDNA samples, and then normalized to the allelic ratio for the corresponding gDNA samples. $P$ values were then determined using a Student's $t$-test for the observed allelic ratios versus the expected ratio of 1.0.

**RNA-seq analysis.** HCASMCs were cultured as described above and stimulated for 1 or 6 h using recombinant human TGF-β1 and PDGF-BB, and total RNA was purified from 5.0E5 cells using the Qiagen miRNeasy kit. RNA libraries were prepared using the Illumina TruSeq library kit as described by the manufacturer. RNA molecules were sequenced using Illumina HiSeq 2500 (2 × 101). Reads contained in raw fastq files were mapped to hg19 using the RNA-seq aligner STAR (v2.4.0i), which processes the data with short run times and yields high numbers of uniquely mapped reads (https://github.com/alexdobin/STAR). Second pass mapping with STAR was then performed using a new index that is created with splice junction information contained in the file SJ.out.tab from the first pass STAR mapping. Reads that have been mapped with STAR second pass mapping algorithm were subsequently counted using the htseq-count script distributed with the HTSeq Python package (https://pypi.python.org/pypi/HTSeq). Differential expression of exons, genes and transcripts were assayed using the DESeq2 R package from Bioconductor (http://bioconductor.org/packages/release/bioc/html/DESeq2.html), which uses negative binomial distribution to estimate dispersion and model differential expression such as to permit biological variability to be different among tested genes (transcripts).

**STARNET gene expression biobank data processing.** Gene expression and genotyping data were obtained from the STARNET database[23]. These consist of RNA-seq gene expression data and genotyping data from nine cardiometabolic tissues from up to 600 CAD patients (determined eligible for the study and consented by the ethical committees of the Karolinska Institutet and Tartu University) that were obtained during coronary artery bypass graft open-heart surgeries. GenomeWideSNP_6 arrays (Affymetrix) were used for genotyping gDNA. Total RNA was isolated from the atherosclerotic arterial wall or internal mammary artery. Gene expression levels were determined using a standard RNA-seq library preparation and sequencing protocol (Illumina HiSeq 2500), followed by normalization of raw read counts to adjust for library size and batch effects. Briefly, samples with <1 million reads were removed, and genes with counts per million <1 in <50% of the samples were also removed. EDAseq was then used to normalize the library size and GC % content, and outliers were removed based on a gender-expression test, covariates were adjusted with linear regression, outliers were again removed and last rank quantile normalization was performed. 'Normalized Expression Counts' represent counts before rank quantile normalization, as shown in Supplementary Fig. 20. Adjusted read counts were subsequently log2-transformed, and the association between genotype and expression was tested using a linear model.

**Validating SNPs in external STARNET CAD eQTL data sets.** eQTLs were called using the MatrixEQTL R package (https://cran.r-project.org/web/packages/MatrixEQTL/index.html) running a linear model and returning all calls with a maximum $P$ value of 0.05. A follow-up conditional analysis was done to test the $P(\text{max SNP} | \text{SNP})$ as well as $P(\text{SNP} | \text{max SNP})$ for each gene. This was calculated by regressing the genotype of SNP_1 onto the residuals of SNP_2 regressed onto the gene expression, where SNP_2 is the SNP being conditioned on (for example,

$P(\text{SNP\_1} | \text{SNP\_2})$). Results were imported into LocusZoom (http://locuszoom.sph.umich.edu/locuszoom/) to generate regional association plots.

**Validating SNPs in public eQTL databases.** In addition to calling eQTLs in STARNET, candidate SNPs were queried in public eQTL databases, including GTEx (http://www.gtexportal.org/home/), seeQTL (http://www.bios.unc.edu/research/genomic_software/seeQTL/), the University of Chicago eQTL Browser (http://eqtl.uchicago.edu/cgi-bin/gbrowse/eqtl/) or the Blood eQTL browser (http://genenetwork.nl/bloodeqtlbrowser/). Queries were performed by rsID or genomic coordinate for candidate SNP, and eQTLs were ranked by $P$ value for all genes. Alternatively, proxy SNPs ($r^2 = 1.0$) were used to report any known eQTLs for the candidate SNPs. Results were imported into LocusZoom (http://locuszoom.sph.umich.edu/locuszoom/) to generate regional association plots.

**Statistical analysis.** All experiments were performed by the investigators blinded to the treatments/conditions during the data collection and analysis, using at least two independent preparations and treatments/conditions in triplicate. The sample sizes for individual experiments were determined based on the power calculations to detect small effects in cultured cells/tissues. R/Bioconductor or GraphPad Prism 6.0 was used for statistical analysis. For enrichment analyses, we used both Fisher's exact test and the cumulative binomial distribution test, as indicated. For comparisons between two groups of equal sample size (and assuming equal variance), an unpaired two-tailed Student's $t$-test was performed or in cases of unequal sample sizes or variance a Welch's unequal variances $t$-test was performed, as indicated. $P$ values <0.05 were considered statistically significant. For multiple comparison testing, two-way analysis of variance accompanied by Tukey's *post hoc* test were used as appropriate.

**Data availability.** All custom scripts have been made available at https://github.com/milospjanic/IntegrativeFunctionalGenomics. Additional modified scripts can be accessed upon request. All sequencing data that support the findings of this study have been deposited in the National Center for Biotechnology Information Gene Expression Omnibus (GEO) and are accessible through the GEO Series accession number GSE72696. All other relevant data are available from the corresponding author on request.

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

## Acknowledgements

We thank all members of the CARDIoGRAMplusC4D consortium for performing the 1000 Genomes analyses and all volunteers for participating in the association studies; William Greenleaf and Anshul Kundaje at Stanford for helpful discussions on data analysis; Michael Snyder, Trupti Kawli, Minyi Shi and Joan Yang at Stanford for assistance with ChIP-seq; Bosh Liu and Kevin Smith for helpful discussions on library QC and sequencing; and Hope Lancero at Stanford for assistance with primary cell culture. We acknowledge support from the following grants: HL109512 (T.Q.), R21 HL120757 (T.Q.), K99 HL125912 (C.L.M.) and a grant from the LeDucq Foundation. The STARNET study was supported by the University of Tartu (SP1GVARENG; J.L.M.B.), the Estonian Research Council (ETF grant no. 8853 (A.R. and J.L.M.B.), the Torsten and Ragnar Söderberg Foundation (C.B.), the Knut and Alice Wallenberg Foundation (C.B.), the Astra-Zeneca Translational Science Centre-Karolinska Institutet

(a joint research program in translational science; J.L.M.B.), the American Heart Association (A14SFRN20840000; E.E.S. and J.L.M.B.) and the National Institute of Health (NIH NHLBI, R01HL125863; J.L.M.B.; NIH NHLBI R01HL71207, E.E.S.). The DNA genotyping and RNA sequencing were in part performed by The SNP&SEQ Technology Platform at Science for Life, The National Genomics Infrastructure in Uppsala and Stockholm with support from Swedish Research Council (VR-RF1), Knut and Alice Wallenberg Foundation and UPPMAX. The funders had no role in the study design, data collection and analysis or preparation of the manuscript.

## Author contribution

C.L.M., M.P. and T.Q. wrote the paper. C.L.M., T.W., T.N., J.D.L., R.K.K., D.M., J.B.K. and O.W. performed the experiments. C.L.M., M.P., A.C., L.P. and E.E.S. analysed the data. A.C., E.E.S., J.L.M.B., C.B., A.R., O.F., L.P., U.H., T.L.A. and S.B.M. contributed reagents/materials/analysis tools. S.B.M. and T.L.A. critically reviewed the manuscript. C.L.M. and T.Q. conceived and designed the experiments. J.L.M.B., A.R. and E.E.S. supervised the STARNET study and analysis. T.Q. and C.L.M. jointly supervised research.

## Additional information

**Competing financial interests:** J.L.M.B. is founder and major shareholder in Clinical Gene Networks AB (CGN) together with A.R. A.R. and E.E.S. are members of the board of directors. CGN has an invested interest in the STARNET biobank and data set.

