## [Peer review file · Nature Communications]

Reviewer #1 (Remarks to the Author)

In this study, Miller and co-workers use an integrative approach to identify causal regulatory variation and genetic mechanisms responsible for associations of variants with CAD in GWAS. An impressive array of technologies were employed, among which, epigenomic and transcriptomic studies in baseline and perturbed cultured human coronary artery smooth muscle cells, and in normal and atherosclerotic human coronary artery tissues. This highlighted a number of variants, which were subsequently studied/validated to some extent. Although the data does not contribute groundbreaking insight into the mechanisms of CAD, the approach for prioritising variants is highly valid and is likely to be emulated in other disciplines. The data appears robust and the manuscript is written clearly.

An important aspect of the work are the epigenetic studies under perturbed conditions. It will be very informative if the authors provide data as to how this has informed the selection of variants as compared to when such perturbation was left out. In other words, how useful / necessary was this perturbation?

Reviewer #2 (Remarks to the Author)

The revised version of this manuscript is improved over the previous version. Just as I had noted in my previous review, the paper offers a tremendous amount of new data that should be of considerable interest to the field. However, I still have concerns about the authors conclusions regarding the enrichment of inflammatory SNPs (and other disease-associated SNPs) in the ATAC-seq peaks from HCASMC cells (Fig 2e). The authors present the results of similar analyses from other cell lines (K562, Glioblastoma cells, and GM12878 cells), and the interpretation of these results is VERY subjective. This needs to be better fleshed out. Specifically, are the open chromatin peaks which harbor the 7 functional SNPs specific to HCASMC cells, or not? Are these sites the same sites that show enrichment of the immune-disorder and schizophrenia SNPs? Or are the immune/schizophrenia SNPs mapping to the same locus, but different sites of open chromatin from the CAD SNPs? Furthermore, it would be nice to know if these regions differ with respect to K27ac levels. This is important, as it has implications for assessing whether or not the CAD SNPs exert cell type specific effects and how they ultimately confer susceptibility to CAD.

Reviewer #3 (Remarks to the Author)

In response to Nature Genetics reviewers comments the authors have responded point by point. Responses:

- 1.1 The authors response to explain the use of TGF- β and PDGF to stimulate the cells prior to genomics analysis is justified with the additions in the main text and figures and the use of appropriate references.
- 1.2 The authors response to explain the use of ATAC-seq and H3K27Ac is satisfactory.
- 1.3 Error is corrected.
- 1.4 Response satisfactory
- 1.5 Response satisfactory (added comments on main text)
- 1.6 Even if selective omission of a rogue sample improved somewhat the separation of PCs it is obvious that there is no clear clustering. Since this is obvious there is not much else that the authors can do other than state the challenging nature of the samples used for readers that are not familiar with these kind of samples.
- 1.7 Response satisfactory (added text and Fisher test p values.
- 1.8 The response of the authors to comments on ASE of SMAD3 is not satisfactory. I found frustrating to navigate the manuscript main text and supplement to identify which SNP was used

as a transcript surrogate for rs17293632 and what is the r^2 or D' value representing the haplotype relationship between the genetic marker and the transcribed SNP. If the authors want to uphold the claim that there is indeed rs17293632 SMAD3 ASE these need to be presented clearly. The use of more inclusive premRNA species does not solve the problem that rs17293632 and the SNP used to assess SMAD3 allelic levels are weakly linked. Since this is an important part of the downstream experimental validation of the genomics analysis I think a more detailed description of the experimental details is needed in the methods section.

1.9 & 1.10 Response satisfactory. From the data presented on fig 6b,c,d it is clear that this is a SNP that disrupts TF binding and is an eSNP for SMAD3 in HACSMC. This might not be statistically significant in cell heterogeneous tissues but the use of single cell types in culture is proving here to be an advantage.

1.11 Response satisfactory. However, as above include a table with the relationship between marker and transcribed surrogate so weak relationships such as (r^2 0.4) are obvious. This is included in the response to comments but has not been incorporated in the manuscript.

1.12 Response satisfactory. Added experimental details

1.13 Response satisfactory. Comments added in discussion.

1.14 Response satisfactory. Correction included

1.15 Response satisfactory

1.16 Response satisfactory

Response to Reviewers Comments – Miller et al., “Integrative functional genomics identifies regulatory mechanisms at coronary artery disease loci”

Reviewer #1 (2)

1.1. Comment: The authors response to explain the use of TGF- β and PDGF to stimulate the cells prior to genomics analysis is justified with the additions in the main text and figures and the use of appropriate references.

Response: N/A

1.2. Comment: The authors response to explain the use of ATAC-seq and H3K27Ac is satisfactory.

Response: N/A

1.3. Comment: Error is corrected.

Response: N/A

1.4. Comment: Response satisfactory

Response: N/A

1.5. Comment: Response satisfactory (added comments on main text)

Response: N/A

1.6. Comment: Even if selective omission of a rogue sample improved somewhat the separation of PCs it is obvious that there is no clear clustering. Since this is obvious there is not much else that the authors can do other than state the challenging nature of the samples used for readers that are not familiar with these kind of samples.

Response: We appreciate the reviewers comments and have thus modified the text to include the additional information on the limited availability and challenge in obtaining clustering data from these human samples.

Results, Page 7, line 20:

It is worth noting that these complex and limited human tissue samples present a challenge to resolve distinct clusters from a standard PCA.

1.7. Comment: Response satisfactory (added text and Fisher test p values.

Response: N/A

1.8. Comment: The response of the authors to comments on ASE of SMAD3 is not satisfactory. I found frustrating to navigate the manuscript main text and supplement to identify which SNP was used as a transcript surrogate for rs17293632 and what is the r^2 or D' value representing the haplotype relationship between the genetic marker and the transcribed SNP. If the authors want to uphold the claim that there is indeed rs17293632 SMAD3 ASE these need to be presented clearly. The use of more inclusive pre-mRNA species does not solve the problem that rs17293632 and the SNP used to assess SMAD3 allelic levels are weakly linked. Since this is an important part of the downstream experimental validation of the genomics analysis I think a more detailed description of the experimental details is needed in the methods section.

Response: We apologize for the confusion, as this point should have been made clearer in the text. We have now included a table as Supplementary Table 16 in the text (also see below), which includes the candidate SNP, transcribed SNP used to detect ASE, as well as the LD information from 1000 Genomes Phase 1 data in Europeans (both r^2 and D' values). As noted in the text, as well as this table, due to the weak linkage between rs17293632 and the proxy transcribed SNP, rs1065080, we measured the allelic imbalance in SMAD3 pre-mRNA levels using the candidate SNP, rs17293632 itself. As we stated in the previous rebuttal, this approach has been successful in deducing ASE for intronic SNPs (Remmers et al. Nat Genet 2010) given that intronic reads are an accurate measure of nascent transcription genome-wide (Gaidatzis et al. Nat Biotech 2015). Just to clarify, these data shown in Fig 6e. have replaced the ASE data using the proxy SNP, rs1065080, as noted in the previous rebuttal. We have now added this justification and experimental details to the results and methods sections.

CAD annotated gene(s)	Candidate CAD SNP	Proxy transcribed SNP	Position (hg19)	MAF (EUR)	Annotation	LD (r^2)	LD (D')
-------------------	-----------------------	-----------------	-----------	------------	--------------	-------------

IL6R	rs7549250	rs2228145	1:154426970	0.36	missense	0.43	0.94
CDKN2B/CDKN2B-AS1	rs1537373	rs3217992	9:22003223	0.40	3'-UTR	0.39	0.75
PDGFD	rs2019090	rs11226057	11:103779025	0.44	3'-UTR	0.39	1.00
SMAD3	rs17293632	rs1065080	15:67457335	0.17	synonymous	0.07	1.00
CCDC97/TGFB1	rs2241718	rs2241718	19:41829606	0.18	3'-UTR	1.00	1.00
LMOD1	rs34091558	rs2820312	1:201869257	0.32	missense	0.94	0.99
TCF21	rs12190287	rs12190287	6:134214525	0.42	3'-UTR	1.00	1.00

Note: Due to weak linkage between *SMAD3* candidate SNP rs17293632 and the transcribed SNP, rs17293632 itself was evaluated for ASE of *SMAD3* pre-mRNA.

Results, page 9, line 23:

Given the weak linkage of a proxy transcript SNP to detect ASE, we measured *SMAD3* pre-mRNA levels using individuals heterozygous at rs17293632 itself. This approach is supported by genome-wide methods correlating intronic reads to changes in transcription{Gaidatzis, 2015 #137}.

Methods, page 21, line 7:

In the case of SNP rs17293632, given the weak linkage between the proxy transcript SNP rs1065080, we used the intronic SNP rs17293632 itself to measure imbalance in the nascent *SMAD3* pre-mRNA as a transcription surrogate{Gaidatzis, 2015 #137}. Another criteria for detecting ASE with these markers was MAF >0.15 in Europeans (as determined from 1000 Genomes data), since this is a limiting factor to obtain sufficient heterozygous individuals from the cohort of HCASMC. Refer to Supplementary Table 16 for further details on these proxy transcribed SNPs.

1.9. & 1.10. Comments: Response satisfactory. From the data presented on fig 6b,c,d it is clear that this is a SNP that disrupts TF binding and is an eSNP for *SMAD3* in HACSMC. This might not be statistically significant in cell heterogeneous tissues but the use of single cell types in culture is proving here to be an advantage.

Response: N/A

1.11. Comment: Response satisfactory. However, as above include a table with the relationship between marker and transcribed surrogate so weak relationships such as (r^2 0.4) are obvious. This is included in the response to comments but has not been incorporated in the manuscript.

Response: We agree that a table would help clarify these details in the manuscript and have now included this table as Supplementary Table 16.

1.12. Comment: Response satisfactory. Added experimental details

Response: N/A

1.13. Comment: Response satisfactory. Comments added in discussion.

Response: N/A

1.14. Comment: Response satisfactory. Correction included

Response: N/A

1.15. Comment: Response satisfactory

Response: N/A

1.16. Comment: Response satisfactory

Response: N/A

Reviewer #3

Comment 3.1. In this study, Miller and co-workers use an integrative approach to identify causal regulatory variation and genetic mechanisms responsible for associations of variants with CAD in GWAS. An impressive array of technologies were employed, among which, epigenomic and transcriptomic studies in baseline and

perturbed cultured human coronary artery smooth muscle cells, and in normal and atherosclerotic human coronary artery tissues. This highlighted a number of variants, which were subsequently studied/validated to some extent. Although the data does not contribute groundbreaking insight into the mechanisms of CAD, the approach for prioritising variants is highly valid and is likely to be emulated in other disciplines. The data appears robust and the manuscript is written clearly.

Response: We appreciate the reviewer's positive comments.

Comment 3.2. *An important aspect of the work are the epigenetic studies under perturbed conditions. It will be very informative if the authors provide data as to how this has informed the selection of variants as compared to when such perturbation was left out. In other words, how useful / necessary was this perturbation?*

Response: We agree that this is important to clarify. In order to fully assess the impact of these perturbations on prioritizing CAD variants, we performed a differential accessibility analysis of ATAC-seq reads under each condition using the edgeR R/bioconductor package. While this software is typically used for gene expression analysis, we adapted it to define differential open chromatin by generating a read count table around the 5,240 candidate CAD SNPs. Using the serum-free condition as the Control and either TGFB, PDGF-BB, or PDGF-DD as the Experimental condition, we implemented a generalized linear model to test for differential accessibility between conditions using a likelihood ratio test to calculate significance at $P < 0.05$. From this analysis we defined 95 variants that resided in differentially accessible chromatin. These results are now included as Supplementary Table 17. Two of these variants were among our top candidates (rs2019090; $P = 0.0010$ and rs73551707; $P = 0.0229$). However, as we noted previously, due to the variance between the different donors (Fig. 1f) and the intrinsic activation of HCASMC when in culture, it may be difficult to identify a large number of differentially accessible regions overall. Nonetheless, we did perform overlaps of the variants using open chromatin regions in HCASMC cultured under these various perturbations. For instance we identified 104, 109, and 105 variants in each of the conditions (TGFB, PDGFBB, and PDGFDD, respectively) compared to 87 variants in the serum-free control condition. After applying functional annotations to these variants using ENCODE and Roadmap data this resulted in 72, 73, and 73 candidate variants, versus 64 in the control condition. In terms of variant selection, these conditions did not significantly influence the final candidate variant list. The more useful "perturbation" turned out to be the ex vivo coronary artery ATAC-seq data, which helped reduce the final candidate variants to 30, and ultimately 26 after applying the annotations. We have now updated Supplementary Fig. 1 to include this information.

Reviewer #4

Comment 4.1. *The revised version of this manuscript is improved over the previous version. Just as I had noted in my previous review, the paper offers a tremendous amount of new data that should be of considerable interest to the field.*

Response: We appreciate the reviewer's positive comments.

Comment 4.2. *However, I still have concerns about the authors conclusions regarding the enrichment of inflammatory SNPs (and other disease-associated SNPs) in the ATAC-seq peaks from HCASMC cells (Fig 2e). The authors present the results of similar analyses from other cell lines (K562, Glioblastoma cells, and GM12878 cells), and the interpretation of these results is VERY subjective. This needs to be better fleshed out. Specifically, are the open chromatin peaks which harbor the 7 functional SNPs specific to HCASMC cells, or not?*

Response:

We agree with the reviewer that we need to better describe these experimental results. Regarding this first question, we should point out that the 7 top candidate SNPs were selected on the basis of integrating multiple sources of functional genomic data in HCASMC, LD information, as well as evidence of function from ENCODE and Roadmap annotations, and not individually on the basis of HCASMC specific regulatory features. This approach has been previously proposed to elucidate functional GWAS variants using ENCODE data (Schaub MA et al Genome Res 2012). Also, there is increasing evidence to argue against the cell-specific role of regulatory variation (Flutre T et al PLoS Genet 2013), which would select for predominately large, highly stable regulatory effects. Nonetheless, to address this question of cellular specificity we have generated HCASMC specific open chromatin regions by subtracting out those regions that co-occur at least once in open chromatin regions from 125 ENCODE cell lines. This generated 6524 HCASMC-specific open chromatin regions that do not occur in the ENCODE datasets. However, our 7 top candidate CAD SNPs were not present

in these HCASMC-specific regions. We then overlapped our 7 top candidate SNPs with each individual ENCODE dataset of open chromatin in 125 cell lines and obtained the following number of overlaps:

CAD SNP	ENCODE open chromatin overlap
rs1537373	3
rs7549250	12
rs2019090	39
rs17293632	42
rs34091558	58
rs73551707	80
rs2241718	87

Therefore, we can conclude that our 7 candidate SNPs are not in HCASMC-specific regions of open chromatin, and are either in a small number of cell lines/tissues (rs1537373, rs7549250) or present in multiple different cell lines/tissues in addition to HCASMC. We have tabulated the ENCODE cell lines/tissues in which these two SNPs appear in open chromatin:

CAD SNP	ENCODE open chromatin overlap (tissue)
rs1537373	Fibroblast (child fibroblast) HeLa (cervical carcinoma) Th2 (primary Th2 T cells)
rs7549250	8988T (pancreas adenocarcinoma) HPDE6-E6E7 (pancreatic duct cells immortalized with E6E7 gene of HPV) HMEC (mammary epithelial cells) HA-H (astrocytes, hippocampal) HA-SC (astrocytes, spinal cord) HCT116 (colorectal carcinoma) HMVEC-DBI (adult blood microvascular endothelial cells, dermal-derived) HMVEC-DBI Neo (neonatal blood microvascular endothelial cells, dermal-derived) HMVEC-D Neo (neonatal microvascular endothelial cells (single donor), dermal-derived) HMVEC-LBI (blood microvascular endothelial cells, lung-derived) HPAEC (pulmonary artery endothelial cells) HRGEC (renal glomerular endothelial cells)

In the case of the open chromatin region covering SNP rs7549250, besides HCASMC this region is accessible in a range of endothelial cells, as well as pancreatic and glial cells, while for rs1537373 this region is accessible in fibroblasts, and also in cervical and T-cells. This indicates that even for the most cell-specific of these 7 SNP/open chromatin regions, cell specificity is not limited to HCASMC or physiologically and phenotypically SMC-related vascular cell types, such as endothelial and fibroblast cells, but instead represents a range of diverse cell types (e.g. astrocytes, pancreatic cells etc), ultimately which may have non-overlapping effects on gene expression.

We also investigated the overlap of our top 7 candidate SNPs with Roadmap Epigenomics open chromatin and histone modification ChIP-seq data from 127 different cells and tissues (many of which are primary tissues not represented in the ENCODE dataset). These tissues are ranked by binomial P-value of enrichment for the SNP overlap versus all GWAS catalog variants as a background. From this analysis we observed that 3 of our top candidate variants (rs1537373, rs17293632, and rs2019090) were enriched in Aorta (in bold), with rs2019090 (PDGFD) enriched in a more limited set of tissues including Aorta. Interestingly all 7 showed significant enrichment for smooth muscle tissues (also in bold) in the GI system (including small intestine, colon, duodenum, esophagus, stomach and rectum). Again, while these regions are not HCASMC specific, they likely represent enhancers that are shared between tissues, related to the developmental origin or physiological function of the tissue composition (e.g. smooth muscle contraction in GI system (peristalsis) versus coronary vasculature (blood flow to heart)). We have summarized this Roadmap data below:

CAD SNP	Roadmap Epigenomics open chromatin and histone mod overlap	Top tissue overlap (binomial P < 0.05 vs all GWAS SNPs)
rs2019090	11	Aorta
rs73551707	42	Colonic mucosa, Rectal smooth muscle , stomach smooth muscle
rs34091558	47	Colonic mucosa, Rectal mucosa, Duodenum smooth muscle , Esophagus
rs1537373	49	Aorta , Primary T-cells, Rectal mucosa, Right atrium, Sigmoid colon
rs2241718	75	ESC, Colonic mucosa, Rectal mucosa, Duodenum smooth muscle , Rectal smooth muscle , Primary B cells, Lung, Right atrium, Primary T-cells, Primary neutrophils from peripheral blood
rs17293632	76	Aorta , Colonic mucosa, Small intestine, Primary T-cells, Rectal smooth muscle , Stomach smooth muscle , Esophagus
rs7549250	82	PBMC, Primary T-cells, Gastric, Rectal smooth muscle , Esophagus, GM12878 lymphoblastoid cells, Ganglion Eminence derived primary cultured neurospheres, Right atrium, K562 Leukemia cells, Primary neutrophils from peripheral blood

Comment 4.3. Are these sites the same sites that show enrichment of the immune-disorder and schizophrenia SNPS?

Response:

To investigate this, we selected GWAS categories for phenotypes: schizophrenia, bipolar, rheumatoid arthritis, ulcerative colitis, Crohns disease, coronary artery and coronary heart, and tested whether the overlaps with HCASMC ATAC-Seq dataset are common among these phenotypes.

HCASMC open chromatin/SNP overlap	Total SNPs per category	GWAS category
3	52	CARDIoGRAMplusC4D
4	89	Coronary_Artery
5	118	Ulcerative colitis
6	130	Coronary_Heart
7	310	Bipolar_disorder
7	189	Crohns_disease
7	156	Schizophrenia
10	204	Rheumatoid_arthritis

However, none of the SNPs nor surrounding open chromatin regions were common among inflammatory- or brain-related phenotypes compared to coronary phenotypes. This indicates that the overlap/significance of these different categories was driven by completely distinct sets of SNP/ HCASMC open chromatin regions.

However, one SNP was common when we intersected brain vs inflammatory categories, specifically Rheumatoid arthritis and Schizophrenia:

Chr	Start	End	Peak intensity	Chr	Start	End	GWAS trait	Gene
chr6	32634396	32634614	57.05	chr6	32634492	32634493	Rheumatoid	TRNAI25

							arthritis	
--	--	--	--	--	--	--	-----------	--

The proxy gene was TRNAI25 (Transfer RNA Isoleucine 25, anticodon AAU).

The list of relevant SNPs and their location, per category, is shown in the following tables:

CARDIoGRAMplusC4D

Chr	Start	End	Peak intensity	Chr	Start	End	GWAS trait	Gene
chr1	109818490	109819040	229.86	chr1	109818530	109818531	CARDIoGRAM	SORT1
chr13	111048706	111050300	760.44	chr13	111049623	111049624	CARDIoGRAM	COL4A1/A2
chr15	91415932	91416553	169.28	chr15	91416550	91416551	CARDIoGRAM	FURIN-FES

Coronary Heart

Chr	Start	End	Peak intensity	Chr	Start	End	GWAS trait	Gene
chr10	102959125	102959634	244.26	chr10	102959339	102959340	Coronary heart disease	CNNM2
chr11	117102558	117103558	377.21	chr11	117103213	117103214	Coronary heart disease	RPS27P19 - PAFAH1B2
chr19	10981346	10982322	658.96	chr19	10981463	10981464	Coronary heart disease	SMARCA4
chr2	170683598	170685463	555.81	chr2	170684313	170684314	Coronary heart disease	HMGB1P4 - LINC01124
chr7	100860457	100861509	347.78	chr7	100860471	100860472	Coronary heart disease	SLC12A9
chr9	22102980	22104014	740.74	chr9	22103814	22103815	Coronary heart disease	CDKN2B-AS1

Coronary Artery

Chr	Start	End	Peak intensity	Chr	Start	End	GWAS trait	Gene
chr10	102959125	102959634	244.26	chr10	102959339	102959340	Coronary artery disease	CNNM2
chr19	11052745	11052996	118.11	chr19	11052925	11052926	Coronary artery disease	SMARCA4
chr6	8000274	8001345	462.62	chr6	8000884	8000885	Coronary artery calcification	BLOC1S5-TXNDC5
chr9	4554948	4555592	305.36	chr9	4555305	4555306	Coronary artery calcification	SLC1A1;S-PATA6L

Schizophrenia

Chr	Start	End	Peak intensity	Chr	Start	End	GWAS trait	Gene
chr1	167933782	167934180	83.81	chr1	167933841	167933842	Schizophrenia	MPC2
chr11	130947053	130947820	276.36	chr11	130947684	130947685	Schizophrenia	SNX19 - NTM
chr15	86440928	86441560	112.75	chr15	86441009	86441010	Schizophrenia	AGBL1
chr18	11494066	11494479	177.66	chr18	11494200	11494201	Schizophrenia	LINC01255
chr5	153160566	153161395	103.17	chr5	153160794	153160795	Schizophrenia	TRNAC32P - GRIA1
chr6	32634396	32634614	57.05	chr6	32634492	32634493	Schizophrenia	TRNAI25
chr7	103764123	103764918	483.30	chr7	103764368	103764369	Schizophrenia	RELN

Bipolar disorder

Chr	Start	End	Peak intensity	Chr	Start	End	GWAS trait	Gene
chr1	30052545	30053069	131.33	chr1	30052867	30052868	Bipolar disorder (mood-incongruent)	PTPRU - MATN1
chr10	60420422	60421406	131.73	chr10	60421370	60421371	Bipolar disorder and major depressive disorder (combined)	ANK3
chr19	1811490	1811851	50.15	chr19	1811604	1811605	Bipolar disorder	ATP8B3
chr20	61531614	61532044	132.27	chr20	61531817	61531818	Bipolar disorder (body mass index interaction)	CDH4
chr22	40600119	40600618	102.54	chr22	40600363	40600364	Bipolar disorder and schizophrenia	MKL1
chr7	2001211	2002026	407.24	chr7	2001797	2001798	Bipolar disorder and schizophrenia	MAD1L1
chrX	14008419 8	14008461 4	168.82	chrX	14008452 4	14008452 5	Bipolar disorder and schizophrenia	HNRNPA3P3 - RPS17P17

Ulcerative colitis

Chr	Start	End	Peak intensity	Chr	Start	End	GWAS trait	Gene
chr1	161509558	161510417	349.04	chr1	161509955	161509956	Ulcerative colitis	FCGR2A
chr12	68110544	68111148	128.94	chr12	68110812	68110813	Ulcerative colitis	HNRNPA1P70 - IFNG
chr5	40410475	40411014	149.37	chr5	40410833	40410834	Ulcerative colitis	LINC00603 - PTGER4
chr7	107862623	107863409	217.10	chr7	107862996	107862997	Ulcerative colitis	PIGCP2 - DLD
chr9	136371692	136372204	345.88	chr9	136371953	136371954	Ulcerative colitis	CARD9

Crohns disease

Chr	Start	End	Peak intensity	Chr	Start	End	GWAS trait	Gene
chr11	57434249	57436313	1055.17	chr11	57435536	57435537	Crohn's disease	SLC43A3 - RTN4RL2
chr13	43883431	43883887	77.01	chr13	43883789	43883790	Crohn's disease	LACC1
chr2	102437746	102438258	147.00	chr2	102437989	102437990	Crohn's disease	IL18RAP
chr5	40410475	40411014	149.37	chr5	40410482	40410483	Crohn's disease	LINC00603 - PTGER4
chr5	150859842	150861003	846.74	chr5	150860514	150860515	Crohn's disease	IRGM - ZNF300
chr9	114803767	114804713	929.47	chr9	114804160	114804161	Crohn's disease	TNFSF15
chr9	136371692	136372204	345.88	chr9	136372044	136372045	Crohn's disease	CARD9

Rheumatoid arthritis

Chr	Start	End	Peak intensity	Chr	Start	End	GWAS trait	Gene
chr1	38157011	38158759	646.87	chr1	38158457	38158458	Rheumatoid arthritis	MIR3659 - TUBB6P1
chr2	191078651	191079977	298.43	chr2	191079016	191079017	Rheumatoid arthritis	STAT4
chr3	27722630	27723290	90.25	chr3	27723132	27723133	Rheumatoid arthritis	EOMES
chr3	58570503	58571601	344.49	chr3	58571114	58571115	Rheumatoid arthritis	FAM107A
chr5	54015313	54016187	232.39	chr5	54015672	54015673	Rheumatoid arthritis	ARL15
chr6	32634396	32634614	57.05	chr6	32634492	32634493	Rheumatoid arthritis	TRNAI25
chr6	106219329	106219917	203.16	chr6	106219660	106219661	Rheumatoid arthritis	ATG5
chr6	167120571	167120982	110.28	chr6	167120802	167120803	Rheumatoid arthritis	CCR6
chr7	92617326	92617725	106.63	chr7	92617430	92617431	Rheumatoid arthritis	CDK6
chr9	34709959	34710384	109.11	chr9	34710263	34710264	Rheumatoid arthritis	CCL21;LOC101929761

Comment 4.4. Or are the immune/schizophrenia SNPs mapping to the same locus, but different sites of open chromatin from the CAD SNPs?

Response:

Immune/schizophrenia SNPs map to different gene loci compared to CAD, which can be seen from the tables shown in response to Comment 4.3.

Comment 4.5. Furthermore, it would be nice to know if these regions differ with respect to K27ac levels. This is important, as it has implications for assessing whether or not the CAD SNPs exert cell type specific effects and how they ultimately confer susceptibility to CAD.

Response:

H3K27ac does not show clear peaks as in transcription factor ChIP-Seq analysis, rather it shows a broad trace, thus here we focused on regions +/- 1000bp surrounding the SNPs and counted the reads with bedtools multicov. When we counted the H3K27ac coverage in a window +/- 1000bp around the 7 candidate CAD SNPs and compared it to the IgG control we obtained the following results. This indicates that these SNPs are surrounded with the H3K27ac modification, however the signal varies among them, with rs17293632 and rs2241718 having the strongest H3K27ac signal represented as fold change normalized to number of reads.

CAD SNP	H3K27ac_rep1	H3K27ac_rep2	H3K27ac_rep3	H3K27ac_rep4	IgG
rs7549250	143	174	112	134	59
rs34091558	327	296	592	244	104
rs2019090	234	225	202	255	91
rs17293632	528	289	520	485	76
rs2241718	11	21	21	31	1
rs73551707	176	146	159	171	111
rs1537373	10	12	27	9	6

Fold change to IgG and normalized to the number of mapped reads:

CAD SNP	H3K27ac_rep1	H3K27ac_rep2	H3K27ac_rep3	H3K27ac_rep4	IgG
rs7549250	2.184	2.822	1.444	2.947	1.000
rs34091558	2.833	2.724	4.329	3.044	1.000
rs2019090	2.317	2.366	1.688	3.636	1.000
rs17293632	6.260	3.639	5.203	8.281	1.000
rs2241718	9.912	20.098	15.970	40.226	1.000
rs73551707	1.429	1.259	1.089	1.999	1.000
rs1537373	1.502	1.914	3.422	1.946	1.000

Next we selected SNPs from the GWAS catalog for schizophrenia, bipolar disorder, rheumatoid arthritis, inflammatory bowel disease, Crohns disease, coronary artery and coronary heart disease, and tested the signal of H3K27ac in +/-1000bp regions. We performed the same analysis as above and obtained normalized fold changes for 4 biological replicate H3K27ac experiments (from different HCASMC donors) and calculated an average value for each SNP. The lead SNPs from different GWAS categories do not differ in their HCASMC H3K27ac signals overall, which is also very low, and that is somewhat expected as lead GWAS SNPs are often located outside histone modification rich regions. However, H3K27ac signals appear to be much higher for the 7 tested SNPs in our study (see Figure below).

To address the reviewer's question we selected SNPs from these GWAS categories that overlap ATAC-Seq open chromatin regions and that were used in the binomial test. As the number of SNPs is not large we decided to calculate the read counts in a larger window (+/-2000bp). We obtained IgG normalized fold changes for 4 H3K27ac experiments in HCASMC and calculated an average value for each SNP. The values are plotted on the graph below for brain-, immune- or CAD-related GWAS categories. We observed a trend that our candidate SNPs and CAD/CHD GWAS SNPs have an increased H3K27ac signal compared to brain and immune GWAS categories.

Finally, we have tested the enrichment of GWAS SNPs in ATAC-Seq open chromatin regions from normal and athero coronary arteries and H3K27ac ChIP-seq active enhancer regions in HCASMC using a binomial test. See table and figures below. The figures have now been added as a new Supplementary Fig. 7. Consistently, we observed that the signal for immuno- and brain-related categories remained high in all three experiments, further supporting our hypothesis of shared pathways between these complex phenotypes.

GWAS trait	Normal artery ATAC	Athero artery ATAC	HCASMC H3K27ac
Schizophrenia	47.182	38.593	138.199
Bipolar disorder	33.730	32.014	132.544
Crohn's disease	35.273	43.354	117.426
Lupus	55.193	43.934	110.833
Coronary Heart	55.536	45.346	103.021

Ulcerative colitis	37.076	20.423	99.239
Asthma	27.469	18.511	93.945
Bone mineral density	17.386	33.216	86.142
Breast cancer	25.796	16.818	85.441
HDL cholesterol	34.772	42.896	85.109
Blood pressure	51.089	58.296	80.863
CARDIoGRAMplusC4D	21.826	10.885	80.291
Multiple sclerosis	18.580	43.567	79.820
Body mass index	32.315	40.628	77.622
Triglycerides	34.866	42.569	70.559
Diastolic blood pressure	41.751	41.544	67.838
Rheumatoid arthritis	61.586	35.726	66.854
Alzheimer's disease	9.905	10.268	66.292
Type 2 diabetes	40.784	35.498	56.892
Liver enzyme levels	0.000	11.776	56.515
Prostate cancer	19.718	29.123	54.578
Type 1 diabetes	10.802	10.529	53.003
Coronary artery calcification	11.581	11.524	45.880
Hypertension	21.120	31.606	45.249
Coronary Artery	21.470	10.831	42.320
Parkinson's disease	19.901	19.610	35.241
LDL cholesterol	28.548	18.972	31.860
Pancreatic cancer	0.000	0.000	29.703
Insulin resistance	12.415	24.251	28.958
Longevity	0.000	0.000	27.967
Primary biliary cirrhosis	0.000	0.000	26.585
C-reactive protein	0.000	0.000	26.394
Lipid metabolism phenotypes	10.639	21.343	26.259
BMI	11.100	10.791	25.645
Fasting plasma glucose	0.000	0.000	23.478
Myocardial infarction	13.447	0.000	22.798
QRS duration	10.825	10.631	18.740
Menarche	0.000	0.000	15.241
Kawasaki disease	11.985	12.349	12.276
FEV1	0.000	0.000	11.633
Celiac disease	10.343	10.378	10.913
Inflammatory biomarkers	0.000	0.000	9.588
Waist-hip ratio	0.000	0.000	0.000

Normal coronary artery ATAC-seq open chromatin regions:

Athero coronary artery ATAC-seq open chromatin regions:

HCASMC H3K27ac ChIP-seq active enhancer regions: